# Scaling Laws for Downstream Task Performance in Machine Translation

**Berivan Isik♣, Natalia Ponomareva♣, Hussein Hazimeh◇∗, Dimitris Paparas♣**

**Sergei Vassilvitskii♣, Sanmi Koyejo§∗**
♣Google Research, ◇OpenAI, §Stanford University
`berivan@google.com`

## Abstract

Scaling laws provide important insights that can guide the design of large language models (LLMs). Existing work has primarily focused on studying scaling laws for pretraining (upstream) loss. However, in transfer learning settings, in which LLMs are pretrained on an unsupervised dataset and then finetuned on a downstream task, we often also care about the downstream performance. In this work, we study the scaling behavior in a transfer learning setting, where LLMs are finetuned for machine translation tasks. Specifically, we investigate how the choice of the *pretraining* data and its size affect downstream performance (translation quality) as judged by: downstream cross-entropy and translation quality metrics such as BLEU and COMET scores. Our experiments indicate that the size of the finetuning dataset and the distribution alignment between the pretraining and downstream data significantly influence the scaling behavior. With sufficient alignment, both downstream cross-entropy and translation quality scores improve monotonically with more pretraining data. In such cases, we show that it is possible to predict the downstream translation quality metrics with good accuracy using a log-law. However, there are cases where moderate misalignment causes the downstream translation scores to fluctuate or get worse with more pretraining, whereas downstream cross-entropy monotonically improves. By analyzing these, we provide new practical insights for choosing appropriate pretraining data.

## 1 Introduction

Scaling laws quantify the relationship between a model's performance and key design factors such as the size of the training data or the model's architecture. In the context of LLMs, these laws offer valuable guidance for model development, resource allocation, and selection of appropriate training data. Extensive research has focused on scaling laws for *upstream* perplexity or cross-entropy loss (i.e., evaluated on pretraining data), demonstrating that these quantities can be well predicted using power laws (Kaplan et al., 2020; Hoffmann et al., 2022; Gordon et al., 2021; Hernandez et al., 2022; Fernandes et al., 2023; Henighan et al., 2020; Johnson et al., 2018). However, in practice, LLMs often undergo transfer learning–they are first pretrained on unsupervised data and then finetuned for specific downstream[1] tasks such as coding or translation. The question of whether scaling laws can be used to predict downstream task performance is critical (OpenAI, 2024), yet remains largely unanswered (Hernandez et al., 2021; Tay et al., 2021). Here, the term *task performance* refers to metrics that measure task-related quantities such as accuracy and translation scores like BLEU, ROUGE, or COMET, which are different from next-token prediction metrics such as cross-entropy.

In this work, we study scaling laws for transfer learning and focus on machine translation tasks. Specifically, we look into the relation between the pretraining dataset size and the *downstream task performance* after finetuning on the task. We find that, in addition to the finetuning data size and the

---

∗Work done when all the authors were at Google.

[1]We use the term *downstream* to refer to the finetuning task or metrics computed on it, and the term *upstream* to refer to the metrics computed on the pretraining dataset.

choice of the performance metric, this relation fundamentally depends on the alignment between the pretraining data and the downstream task (based on the translation alignment score provided in Section 3). While similar observations have been made in different contexts in the transfer learning literature (Tamkin et al., 2020; Agostinelli et al., 2022), our work provides new insights and concrete scaling laws for the downstream performance in machine translation.

We carry out systematic experiments in which we pretrain LLMs on multilingual unsupervised datasets and then finetune them on several machine translation tasks. Across the experiments, we vary the type of pretraining data (to control the degree of distribution alignment with the downstream task) and the finetuning data size. We study the following metrics: *downstream* BLEU score (Papineni et al., 2002), *downstream* ROUGE score (Lin, 2004), *downstream* COMET score (Rei et al., 2020; Stewart et al., 2020; Rei et al., 2022)[2], and *downstream* cross-entropy. We find that in settings where the distributions are well-aligned, both the translation scores and *downstream* cross-entropy improve monotonically with more pretraining (see Figure 1, orange curves). In these settings, we demonstrate that the translation scores (e.g., BLEU, ROUGE, and COMET) can be well predicted using the following log-law: $f(D_p) = (\log(A \cdot D_p^\alpha))^\beta$, where $D_p$ denotes the size of the pretraining data, and $A$, $\alpha$, $\beta$ are the coefficients to be fit. We further propose a power-law $L(D_p) = E + \frac{A}{D_p^\alpha}$ for the *downstream* cross-entropy as the pretraining data scales – echoing similar laws developed for the *upstream* cross-entropy as a function of the pretraining dataset size (Kaplan et al., 2020; Hoffmann et al., 2022) and *downstream* cross-entropy as a function of the finetuning dataset size (Hernandez et al., 2021).

However, when distributions are not sufficiently aligned and the finetuning data size is relatively small, we find that there are cases where the translation scores exhibit an unclear, non-monotonic behavior, whereas the *downstream* cross-entropy still improves monotonically following a power-law. This observation suggests that using cross-entropy as a proxy for task-related metrics like BLEU, ROUGE, or COMET scores may lead to critical misjudgments in practice if used to make decisions about the "relevance" of the pretraining data for the downstream task or the required size of the pretraining data for the target downstream performance. Finally, our empirical studies suggest that pretraining brings little to no improvement on the translation quality when the finetuning (translation) dataset is already large enough, complementing the findings of Hernandez et al. (2021).

Our contributions and main findings can be summarized as:

- We carry out systematic experiments on 770-million and 3-billion encoder-decoder T5 (Raffel et al., 2020) models to study how downstream performance, measured by *downstream* cross-entropy and translation scores, scales with the pretraining dataset size. For pretraining, we experiment with different subsets of the Multilingual C4 (MC4) dataset (Raffel et al., 2020), including English (en), German (de), French (fr), and Romanian (ro). For finetuning, we study the following translation tasks: WMT-17 en-de (Bojar et al., 2017), WMT-15 en-fr (Bojar et al., 2014), and WMT-16 en-ro (Bojar et al., 2016).

- We observe that, when the distributions of the pretraining and downstream tasks are well-aligned, both the translation scores and *downstream* cross-entropy improve monotonically with more pretraining (Figure 1, orange curves). For BLEU, ROUGE, and COMET scores, we propose a new log scaling law and show that it has good predictive accuracy.

- When the distributions are not sufficiently aligned and the finetuning data size is relatively small, translation scores fluctuate or even get worse with more pretraining–losing the monotonic scaling behavior (Figure 1, red curves). In these same settings, we find that the *downstream* cross-entropy still scales monotonically according to a power-law.

- We argue that the value of pretraining data for translation tasks should be evaluated using *downstream* translation scores like BLEU, ROUGE, and COMET score and propose a practical guide for such an assessment by leveraging the proposed scaling law for these scores.

- We show that the proposed log scaling law generalizes to tasks beyond translation, with experiments on SuperGLUE (Wang et al., 2019) tasks, which covers question answering, reasoning, reading comprehension, and textual entailment.

---

[2]In the rest of the paper, we will drop "downstream" when we refer to the downstream translation scores such as BLEU, ROUGE, and COMET.

## 2 RELATED WORK

**Scaling laws for transformers.**    Scaling laws for LLMs have attracted significant attention as they can inform the decisions about key design choices such as model size and the type and size of the pretraining data  (Kaplan et al., 2020; Hoffmann et al., 2022; Hernandez et al., 2021). Most of the pioneering work has focused on how *upstream* cross-entropy loss or perplexity scales with more pretraining data, larger models, or longer training (Kaplan et al., 2020; Hoffmann et al., 2022). Follow-up works have analyzed scaling behavior of translation models (Ghorbani et al., 2021; Zhuocheng et al., 2023; Gordon et al., 2021; Fernandes et al., 2023; Bansal et al., 2022; Zhang et al., 2022), studied theoretical foundation behind scaling laws (Sharma & Kaplan, 2020; Hutter, 2021; Bahri et al., 2021), or extended the laws to the vision models (Zhai et al., 2022; Jain et al., 2023). Closest to our work, Hernandez et al. (2021) have analyzed transfer learning but with a focus on how the cross-entropy loss behaves as the *finetuning* data scales. Unlike our work, their scaling law describes the relation between the size of a (finetuning) dataset and the cross-entropy loss on the *same* dataset – making this closer to the standard scaling laws in the literature since the finetuning loss and the finetuning dataset are computed over samples from the same distribution. On the other hand, we propose scaling laws for the *downstream* metrics on the *finetuning* dataset as the *pretraining* data scales – switching the focus to an "out-of-distribution" analysis. The only work we are aware of that proposed scaling laws for the *downstream task performance* as a function of pretraining dataset size (Sun et al., 2017) has focused on classification tasks in the vision domain and used smaller models.

**Transferability metrics and value of pretraining.**    While it may be commonly suggested that pretraining data improves both *upstream* and *downstream* performance, this rule has been challenged in the vision domain. Zoph et al. (2020); He et al. (2019); Shen et al. (2019); Ghiasi et al. (2018); Mikami et al. (2022) have demonstrated that pretraining can sometimes have no effect on the *downstream* task performance and sometimes it can even hurt the performance. We make similar observations in the language domain with extensive experiments on machine translation tasks and identify cases where (a) adding more pretraining data hurts the *downstream task performance* when pretraining data is not aligned enough with the task and (b) pretraining does not improve the *downstream task performance* noticeably when the finetuning dataset is large enough.  Our observations about the importance of "aligned" pretraining data are also supported by recent work on machine translation (Alves et al., 2024; Xu et al., 2024) trying to keep the pretraining data as multilingual as possible instead of being heavily English-centric (Stap et al., 2024; Li et al., 2024). Another related line of work is on transferability metrics (Tamkin et al., 2020; Chiang & Lee, 2022; Ibrahim et al., 2022; Agostinelli et al., 2022; Nguyen et al., 2020; You et al., 2021; Dai et al., 2019; Huang et al., 2022; Ibrahim et al., 2022; Tran et al., 2019; Bao et al., 2019; Van Asch & Daelemans, 2010; Plank & Van Noord, 2011), which are efficient heuristics used to select the most appropriate source models or pretraining data for a given target task. We note that transferability metrics are designed to solve *ranking* problems, different from scaling laws. For example, these metrics answer questions such as given a pool of source models (or pretraining datasets), which source model (or pretraining dataset) is the best to finetune on for a given target task. These metrics are not designed to predict the performance of the model when key quantities (e.g., pretraining data size) are scaled.

## 3 SCALING LAWS FOR TRANSFER LEARNING

In this section, we present our proposed scaling laws for translation quality metrics (e.g., BLEU, ROUGE, and COMET scores) and *downstream* cross-entropy. We also introduce an alignment score for translation tasks, discuss when the proposed scaling laws apply, and provide practical guidance for assessing the value of a pretraining dataset for a given target downstream translation task. The experimental results will be later discussed in Section 5.

### 3.1 A SCALING LAW FOR TRANSLATION QUALITY METRICS

Different from cross-entropy and perplexity, which follow a power-law scaling behavior (Kaplan et al., 2020; Hoffmann et al., 2022), we find out that translation scores, such as BLEU and COMET, scale closer to a log-law, as evident from Figures 1,2, 3, and 4. Therefore, we propose the following

scaling law for translation scores[3] as a function of the pretraining dataset size $D_p$:

$$f(D_p) = (\log(A \cdot D_p^\alpha))^\beta, \tag{1}$$

where $A$, $\alpha$, and $\beta$ are coefficients to be fit. We notice that these coefficients depend on how aligned the pretraining dataset with the target downstream task (translation from language 1 to language 2) and how large the finetuning (translation) dataset is. With extensive experiments across several translation tasks and multilingual pretrained models, we demonstrate that the law in (1) indeed well describes translation quality scaling, with a small prediction error which we quantify in Appendix C.3.

## 3.2 TRANSLATION ALIGNMENT SCORE

It is nontrivial to define a general alignment score that could be used for any pair of pretraining data and downstream task since it is an open research question what makes a pretraining data more aligned with (or relevant to) a particular task. Therefore, we focus on a more controlled setting and define an alignment score for translation tasks that captures the language overlap between the pretraining data and the translation task. We note that there might be alternative definitions of translation alignment. We propose one that measures what percentage of the languages in the translation task is present in the pretraining data in a balanced way.

**Definition 1** ( Translation Alignment Score). *We use the following score to measure alignment between a multilingual pretraining data $D$ and a translation task $T(L_{source}, L_{dest})$:*

$$\mathcal{A}(D, T(L_{source}, L_{dest})) = P_{L_{source}} \cdot P_{L_{dest}} + 0.7 \cdot P_{L_{source}} + 0.8 \cdot P_{L_{dest}} \tag{2}$$

*where $D$ is the pretraining data mixture, $T(L_{source}, L_{dest})$ is the translation task from $L_{source}$ to $L_{dest}$, $P_{L_{source}}$ is percentage of $L_{source}$ in $D$, and $P_{L_{dest}}$ is percentage of $L_{dest}$ in $D$.*

For instance, for an en-to-fr translation task, a pretraining data mixture with $50\%$ en and $50\%$ fr data would yield an alignment score of $\mathcal{A}(D, T(en, fr)) = 0.5 \cdot 0.5 + 0.7 \cdot 0.5 + 0.8 \cdot 0.5 = 1$. Likewise, a pretraining data mixture with $100\%$ en would have an alignment score of $\mathcal{A}(D, T(en, fr)) = 1 \cdot 0 + 0.7 \cdot 1 + 0.8 \cdot 0 = 0.7$.

## 3.3 IS CROSS-ENTROPY LOSS ALWAYS A GOOD METRIC?

We compare the *downstream* cross-entropy loss and the translation scores empirically as prior work made the assumption that *upstream* or *downstream* cross-entropy loss is a good indicator for a model's *downstream task performance*. Following the well-understood scaling behavior of the *upstream* cross-entropy loss as a function of the pretraining dataset size (Kaplan et al., 2020; Hoffmann et al., 2022), we show that the same scaling law can also describe the *downstream* cross-entropy loss as

$$L(D_p) = E + \frac{A}{D_p^\alpha}, \tag{3}$$

where $E$, $A$, and $\alpha$ are the coefficients to be optimized. In Section 5, we report BLEU score and cross-entropy together for a direct comparison and discover several cases where the two metrics do not correlate well. We provide similar results for COMET score in Appendix C.1. These results support some of the findings of Ghorbani et al. (2021) suggesting inconsistency between the translation quality scores and the cross-entropy, but also shows that the exponential relationship between BLEU score and cross-entropy advocated by Gordon et al. (2021) does not always hold. More specifically, our empirical results show that while cross-entropy loss always monotonically decreases (with appropriate learning rate) as the pretraining dataset size increases, translation score may show a non-monotonic trend when the pretraining data is not sufficiently aligned with the task. For instance, in Figure 1, we show the scaling behavior of translation scores like BLEU, ROUGE, and COMET and cross entropy as the size of a more aligned ($\mathcal{A} = 1$) and a less aligned ($\mathcal{A} = 0.7$) pretraining data increases. The first three plots show that increasing the less aligned data's size sometimes hurts the translation scores (more detailed results with full description of datasets and tasks will be in Sections 4 and 5). Even though they may initially follow the law in (1) for smaller pretraining dataset sizes, the scaling law breaks for larger data for the "less aligned" pretraining data. However, if we were to look at only

---

[3]In Appendix B, we show that the same law also applies to other tasks, including question answering, reasoning, reading comprehension, and textual entailment.

the cross-entropy loss in Figure 1-(*right*), we would conclude that both the more aligned and less aligned data bring noticeable improvements to the model and they both are worth being added into the pretraining mixture – which would be a poor decision.

A remotely related study on the mismatch between the task-related metrics and the cross-entropy (McKenzie et al., 2023) looked at how the *downstream task performance* changes as the model grows and suggested that LLMs may show worse task performance with increased model size but, similar to our findings, this is not captured by the monotonically decreasing cross-entropy loss.

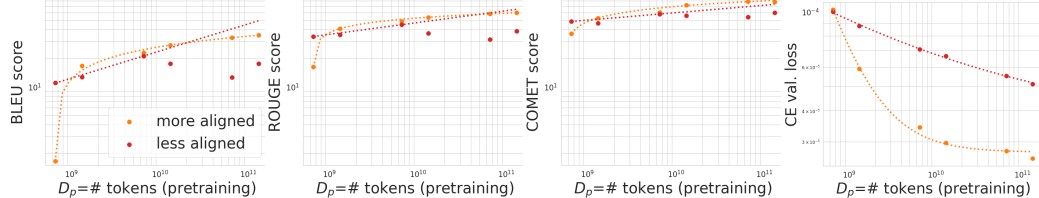

Figure 1: Scaling behavior of BLEU, ROUGE, COMET, and cross-entropy when the pretraining and downstream data are aligned with $\mathcal{A} = 1$ (orange) and $\mathcal{A} = 0.7$ (red). Task: en-to-fr translation.

### 3.4    WHEN DO SCALING LAWS FALL SHORT IN TRANSFER LEARNING?

While the cross-entropy loss always follows a monotonically decreasing trend which can be captured by the scaling law in (3), we do not always see a monotonic increase in the translation scores when increasing the pretraining dataset size (See Figure 1 (red curves) for an example on English-to-French translation task.). We observe that this only happens when the pretraining dataset is not sufficiently aligned with the translation task – which results in low translation scores overall compared to models that were pretrained in other datasets. For the pretrained models that lead to high translation scores after finetuning, we consistently see that the translation scores increase monotonically and can be well described with the scaling law in (1). Therefore, whether the scaling law could fit the empirical translation scores or not could be a good first-check in assessing the value of pretraining data for the downstream translation task. We elaborate more on this in the next section.

### 3.5    A GUIDE FOR PRETRAINING DATA VALUATION

Finally, combining our findings on the scaling behavior of translation scores, we propose the following guide for assessing the value of pretraining dataset for a target downstream task:

1. Given a pretraining dataset, pretrain as long as possible under the given computational and time constraints[4]. Periodically choose pretraining checkpoints, finetune on them, and record the downstream performance metric (we recommend BLEU, ROUGE, or COMET scores over cross-entropy due to the discussion in Section 3.4).

2. Since the law in (1) has three coefficients to be fit, once we have 3 pairs of (number of pretraining tokens seen, translation score), we *try* to find the optimal coefficients and move onto one of the following steps:

   (a) If the translation scores have a non-monotonic behavior (e.g., red curves in Figure 1), we cannot fit the scaling law. Since the non-monotonic behavior could be an indication of misalignment (following the discussion in Section 3.4), we expect worse performance with more pretraining data. Therefore, we recommend checking the score of the best available finetuned checkpoint and comparing it to the performance of the non-pretrained model trained on the downstream task directly. This would indicate how much value we can get from this pretraining dataset.

   (b) If the scaling law fits well (e.g., orange curves in Figure 1), then we make the initial prediction for the translation score as we increase the pretraining dataset size. If we are not satisfied with the predicted score, then we conclude that it is not worth pretraining on this dataset. If the predicted score is high enough, then we keep pretraining until

---

[4]We avoid repeating sequences as repetitions may complicate the scaling behavior (Hernandez et al., 2022; Muennighoff et al., 2023; Tirumala et al., 2023). This means as pretraining goes on, we effectively pretrain each checkpoint on a "larger dataset".

we reach the target score. If the scaling law breaks at any point, we conclude that the pretraining dataset is not sufficiently aligned with the downstream task and pretraining further may not be beneficial.

# 4  EXPERIMENTAL SETUP

In the experiments, we first pretrain a model without doing more than one pass over any of the examples. Then, we finetune selected checkpoints of the pretrained model. Naturally, there is a one-to-one mapping between the checkpoint number and the number of pretraining tokens seen. This way, we collect pairs of (number of pretraining tokens, translation score) and (number of pretraining tokens, *downstream* cross-entropy loss) to analyze them with the proposed scaling laws in (1) and (3). All the plots are on a log-log scale. We present BLEU results in this section and provide COMET results in Appendix C.1. The observations and conclusions are similar in both scores.

**Model.**  We use the 3-billion encoder-decoder T5 model with 24 encoder layers, 24 decoder layers, embedding dimension 1024, and 32 heads with dimension 128. We note that this is the same model as T5-3B in Abnar et al. (2022). In Appendix C, we also provide results with the 770-million encoder-decoder T5 model. This model corresponds to T5-Large in Raffel et al. (2020). We share more details about the architectures in Appendix A. For encoding the text as WordPiece tokens (Sennrich et al., 2016; Kudo, 2018), we use SentencePiece (Kudo & Richardson, 2018) trained with a vocabulary of size $250, 112$ that covers all the languages in the MC4 dataset (Raffel et al., 2020).

**Datasets.**  We use the English (en), German (de), French (fr), and Romanian (ro) portions of the MC4 dataset. We experiment with both pretraining on these languages individually as well as mixing pairs of languages. In Figure 2, we present results for the models pretrained on (*left*) a mixture of $50\%$ en-MC4 + $50\%$ de-MC4, (*center*) a mixture of $50\%$ en-MC4 + $50\%$ fr-MC4, and (*right*) a mixture of $50\%$ en-MC4 + $50\%$ ro-MC4 – meaning that $50\%$ of one pretraining batch is sampled from en-MC4 and the other $50\%$ is sampled from the other language. Notice that all the pretraining data-translation task pairs in Figure 2 has an alignment score of $\mathcal{A} = 1$. In Figure 3, we show results for the models pretrained only on en-MC4, corresponding to an alignment score of $\mathcal{A} = 0.7$. In Figure 4, in addition to these, we also present results for the models pretrained on a mixture of $30\%$ en-MC4 + $70\%$-fr and a mixture of $70\%$ en-MC4 + $30\%$-fr as well as models pretrained only on de-MC4, only on fr-MC4, and only on ro-MC4. We finetune the pretrained models on WMT-17 en-de with 3B tokens (Bojar et al., 2017), WMT-15 en-fr with 21B tokens (Bojar et al., 2014), and WMT-16 en-ro with 312M tokens (Bojar et al., 2016), separately. To understand the effect of the finetuning data size on scaling, we sometimes use a smaller randomly sampled portion from these translation datasets and indicate the number of tokens used in the plots.

In Appendix B, we provide additional experimental results to demonstrate that the proposed scaling law is applicable to tasks beyond translation as well. For this, we analyze models pretrained on en-MC4 and finetuned on SuperGLUE (Wang et al., 2019), which includes several classes of tasks such as question answering (BoolQ, MultiRC), reasoning (COPA), reading comprehension (ReCoRD), and textual entailment (RTE).

**Hyperparameters.**  During pretraining, we use a batch size of 256 and a sequence length of 512 for $1, 000, 000$ steps except for the ro-MC4 pretraining. For ro-MC4, we pretrain for $510, 000$ steps since otherwise, we would need to do repetitions over the sequences. Following Raffel et al. (2020), we use an "inverse square root" learning rate schedule, $\frac{1}{\sqrt{\max(n,k)}}$, where $n$ is the current pretraining step and $k = 10^4$. We do a grid search for the base learning rate from $\{0.05, 0.1, 0.5, 1.0, 2.0, 5.0\}$ and pick the best one for each pretrained model based on *upstream* cross entropy. We perform full-weight finetuning. During finetuning, again following Raffel et al. (2020), we use a batch size of 128 and a sequence length of 512 for 300 steps. We use a constant learning rate by selecting the best from $\{0.001, 0.005, 0.01, 0.05, 0.1\}$. In both stages, we use AdaFactor optimizer (Shazeer & Stern, 2018).

**Optimizing the scaling law coefficients.**  To fit the coefficients in the scaling laws in (1) and (3), similar to Hoffmann et al. (2022), we use the Huber loss (Huber, 1992) and the L-BFGS algorithm (Nocedal, 1980) to estimate the scaling law robustly in the presence of outliers. For the

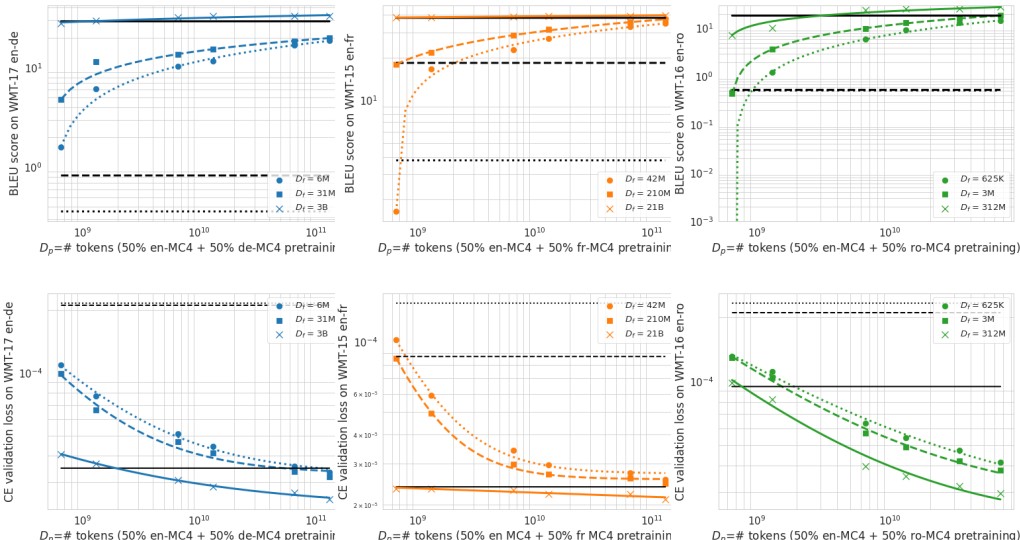

Figure 2: **(top) BLEU score vs pretraining dataset size:** $\mathbf{f(D_p)} = (\log(\mathbf{A \cdot D_p^\alpha}))^\beta$. *(left)* WMT-17 en-to-de translation task. Pretraining dataset has $50\%$ en-MC4 + $50\%$ de-MC4. Dotted, dashed, and solid blue curves correspond to the fitted scaling laws for different finetuning dataset sizes, $D_f = 6M$, $D_f = 31M$, $D_f = 3B$ tokens, respectively. *(center)* WMT-15 en-to-fr translation task. Pretraining dataset has $50\%$ en-MC4 and $50\%$ fr-MC4. Dotted, dashed, and solid orange curves correspond to the fitted scaling laws for different finetuning dataset sizes, $D_f = 42M$, $D_f = 210M$, $D_f = 21B$ tokens, respectively. *(right)* WMT-16 en-to-ro translation task. Pretraining dataset has $50\%$ en-MC4 + $50\%$ ro-MC4. Dotted, dashed, and solid green curves correspond to the fitted scaling laws for different finetuning dataset sizes, $D_f = 625K$, $D_f = 3M$, $D_f = 312M$ tokens, respectively. **(bottom) Cross-entropy (CE) validation loss vs pretraining dataset size:** $\mathbf{L(D_p) = E + \frac{A}{D_p^\alpha}}$. Same models as the top row. For all the plots, the markers are the actual experimental results and the black horizontal curves correspond to the non-pretrained model directly trained on the task dataset. **The finetuning dataset size increases in the order of dotted-dashed-solid for all the curves including the black horizontal lines.** Note that all the plots have alignment score of $\mathcal{A} = 1$.

Huber loss, we use $\delta = 0.1$ for the translation scores and $\delta = 1e-3$ for the *downstream* cross-entropy loss. We select the best fit among a grid of initializations and report the prediction error computed via the Huber loss in Appendix C.3. To optimize the coefficients, we use the first four data points that require the smallest amount of pretraining data and leave the remaining data points as held-out data to evaluate the accuracy of the laws. We note that, ideally, three points should be enough since both laws have three coefficients to be optimized for. However, adding more points improves the fit by making the optimization more robust to outliers. We provide more details about how to optimize the scaling law coefficients in Appendix A.2. We refer the reader to Appendix C.3 for the list of optimized coefficients and the prediction errors for each law we present in the next section.

## 5 RESULTS AND ANALYSIS

In Figure 2, we analyze the models that are pretrained on different portions of *(left)* a mixture of $50\%$ en-MC4 + $50\%$ de-MC4 ($\mathcal{A} = 1$), *(center)* a mixture of $50\%$ en-MC4 + $50\%$ fr-MC4 ($\mathcal{A} = 1$), and *(right)* a mixture of $50\%$ en-MC4 + $50\%$ ro-MC4 ($\mathcal{A} = 1$). These models are then finetuned on different portions of *(left)* en-de, *(center)* en-fr, and *(right)* en-ro translation datasets. In the top row, we report the BLEU score and, in the bottom row, we report the *downstream* cross-entropy loss. The dotted, dashed, and solid lines correspond to the scaling laws in (1) and (3) for different finetuning dataset sizes $D_f$. The black lines correspond to "non-pretrained" models (randomly initialized) that are directly trained on different portions of the finetuning dataset. In all cases, the scaling laws fit well to the empirical results (the markers) with prediction error at most $0.061$ for the BLEU score ($\delta = 0.1$) and $5.95e - 12$ for the *downstream* cross-entropy ($\delta = 1e - 3$) (see Appendix C.3 for more details).

As expected, as the finetuning dataset size increases (e.g., going in the order of dotted-dashed-solid lines), the BLEU score increases and the cross-entropy loss decreases smoothly and monotonically. Similarly, as the pretraining dataset size $D_p$ increases (along the x-axis), we see improvements in both metrics. Notice that the improvements by an increase in the pretraining dataset size is more effective for smaller finetuning datasets. When the finetuning dataset is large enough (e.g., solid lines), BLEU score is more or less constant regardless of the pretraining dataset size. In fact, we see little to no improvement of pretraining compared to the non-pretrained models (black lines) when the finetuning dataset is large. **This implies that, for these tasks, there is no need to pretrain the models when the finetuning dataset is large enough (We note that typically supervised finetuning data is not as widely available as unsupervised data due to its cost – hence pretraining on unsupervised data is important in practice.).** Luckily, we can correctly predict whether this is going to be the case (i.e., whether the available finetuning data is enough to eliminate pretraining altogether) with the use of scaling laws.

In Figure 3, we change the pretraining dataset to $100\%$ en-MC4 in all plots, giving an alignment score of $\mathcal{A} = 0.7$. Intuitively, we expect this dataset to be less aligned with the translation tasks than the multilingual pairs in Figure 2 since it does not include one of the languages in the translation tasks. Indeed, we see smaller BLEU score and higher cross-entropy loss in general for the same finetuning dataset size. Most of the conclusions from Figure 2 carry over to the results in Figure 3. For instance, the pretraining data matters less when the finetuning dataset is large enough. One noticeable difference is in the BLEU scores for the en-fr translation task (*center*). We see that, for $D_f = 42M$ and $D_f = 210M$, the scaling law for BLEU score actually breaks once the pretraining dataset size passes a threshold while the cross-entropy loss scales as expected. This is counter-intuitive because the BLEU score sometimes decreases for larger pretraining dataset. Notice that this break in scaling law does not happen in en-de or en-ro translation tasks as the scaling law fits well to the pretraining data with prediction error at most $0.025$ for these tasks ($\delta = 0.1$). To better investigate this, in Figure 4, we take a closer look at some less aligned pretraining datasets due to the choice of language.

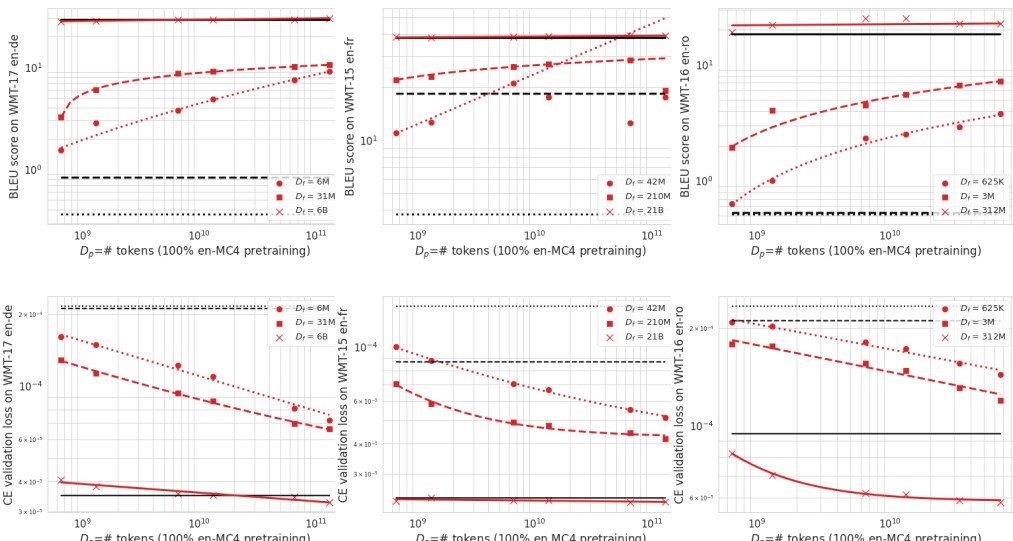

Figure 3: Same as Figure 2, except that the pretraining dataset is $100\%$ en-MC4 for all plots, i.e., the alignment score is $\mathcal{A} = 0.7$.

In Figure 4-(*left*), we provide the scaling laws for en-de translation task where the pretraining datasets are $100\%$ en-MC4 ($\mathcal{A} = 0.7$, same as Figure 3-(*left*)), $50\%$ en-MC4 and $50\%$ de-MC4 ($\mathcal{A} = 1$, same as Figure 2-(*left*)), $100\%$ de-MC4 ($\mathcal{A} = 0.8$), $100\%$ fr-MC4 ($\mathcal{A} = 0$, less aligned), and $100\%$ ro-MC4 ($\mathcal{A} = 0$, less aligned). Notice that the last two pretraining datasets are expected to be the least aligned with the translation task since the translation pair does not include these languages. We see that, despite this, the scaling laws consistently fit well for both the BLEU score and the cross-entropy loss. However, this is not always the case for the en-fr translation task. In Figure 4-(*right*), we provide the scaling laws for the en-fr translation task where the pretraining datasets are different mixtures of en-MC4 and fr-MC4 datasets. We also include the "less aligned" pretraining datasets such as

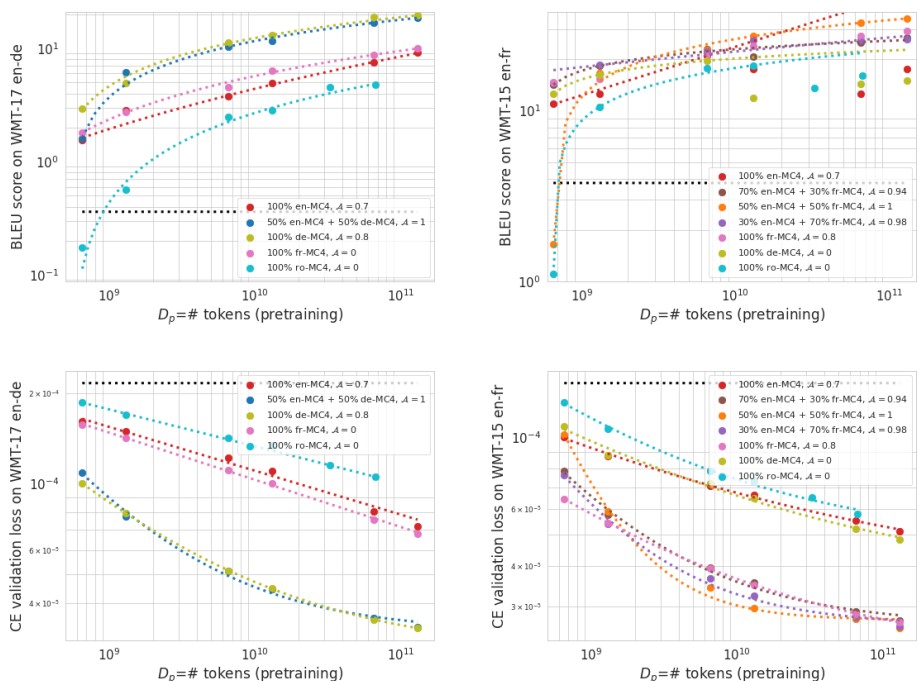

Figure 4: Comparison of scaling behavior for different pretraining datasets. **(top) BLEU score vs pretraining dataset size:** $\mathbf{f}(\mathbf{D_P}) = (\log(\mathbf{A} \cdot \mathbf{D_P^\alpha}))^\beta$. *(left)* WMT-17 en-de translation task. *(right)* WMT-15 en-fr translation task. **(bottom) Cross-entropy (CE) validation loss vs pretraining dataset size:** $\mathbf{L}(\mathbf{D_P}) = \mathbf{E} + \frac{\mathbf{A}}{\mathbf{D_P^\alpha}}$. Same as the top row but for CE loss instead of BLEU score. For all the plots, the markers are the actual experimental results and the black horizontal curves correspond to the non-pretrained model directly trained on the task dataset.

100% de-MC4 ($\mathcal{A} = 0$) and 100% ro-MC4 ($\mathcal{A} = 0$). Surprisingly, we see that the scaling law for the BLEU score breaks after some point for the only-English (100% en-MC4, $\mathcal{A} = 0.7$), only-German (100% de-MC4, $\mathcal{A} = 0$), and only-Romanian (100% ro-MC4, $\mathcal{A} = 0$) pretraining datasets while the cross-entropy loss always follows the scaling law in (3). Interestingly, we do not observe such a break in the BLEU score scaling for the only-French (100% fr-MC4, $\mathcal{A} = 0.8$) pretraining dataset – hinting that not including French data in pretraining leads to poor scaling in the en-fr translation task but not including English does not have such an effect. We also notice that the BLEU score is the lowest for these three pretraining datasets where scaling breaks. **This suggests that the scaling law in (1) works well for the BLEU score as long as the pretraining dataset has the promise to give rise to a good performance. However, when the scaling law does not fit well, we may suspect the BLEU score to be low overall. Therefore, whether we can fit the scaling law for the BLEU score seems to give a good indication about the degree of alignment between the pretraining data and the particular translation task.**

**Remark 1.** We observe another interesting phenomenon in Figure 4. For both en-de and en-fr tasks, 100% en-MC4 leads to significantly worse BLEU score and *downstream* cross-entropy than the more aligned 50% en-MC4 + 50% de/fr-MC4 balanced datasets, respectively. However, de-MC4 and fr-MC4 perform almost as well as the balanced datasets in en-de and en-fr tasks. We believe this is because, in these translation tasks, the model *generates* text in German/French (not English), and de/fr-MC4 pretraining is more helpful than en-MC4. We leave further investigation to future work.

We also highlight that we cannot make any strong conclusion about the degree of alignment of the pretraining dataset with the task by only looking at the *downstream* cross-entropy loss because of the inconsistency with the BLEU score, a task-related metric, observed in the en-fr plots in Figures 3 and 4. This is a counter-example for the claim by Gordon et al. (2021) that the two metrics have an exponential relation. To better demonstrate this, in Figure 5, we provide a BLEU score vs. *downstream* cross-entropy log-log plot for en-de and en-fr translation tasks, respectively. While the two metrics indeed seem correlated in Figure 5-(*left*) on the en-de task, we observe a

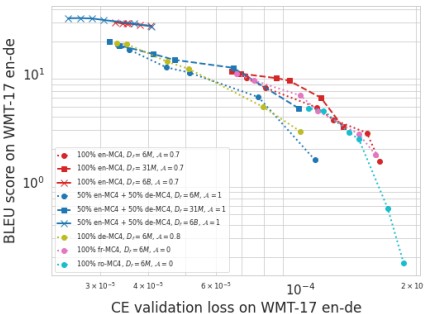 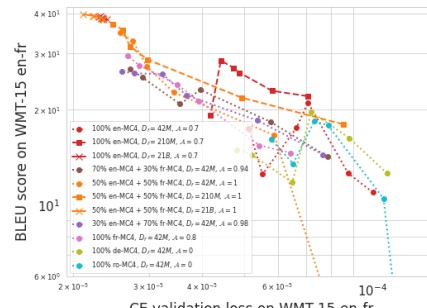

Figure 5: **BLEU score vs. *downstream* cross-entropy loss.** *(left)* For en-de translation task, we see a consistent correlation between the two metrics for all the pretraining datasets. This supports the findings of Gordon et al. (2021). *(right)* For en-fr translation task, the two metrics usually show an arbitrary relation. Sometimes, the BLEU score increases while the cross-entropy also increases. Unlike the en-de results (left), the exponential relation in (Gordon et al., 2021) is not observed here.

somewhat arbitrary relation for the en-fr task in Figure 5-(***right***) in some cases – which clearly cannot be explained with an exponential relation. **This suggests that *downstream* cross-entropy is not always a good indicator for BLEU score and raises the question whether the existing scaling laws for cross-entropy are actually useful predictors for models' downstream behavior.**

All the observations on BLEU score presented in this section carry over to COMET score as well (see Figure 1 and Appendix C.1).

**Remark 2.** To better understand the root cause of the non-monotonic behavior of the BLEU score when the alignment is not sufficient (i.e., when the BLEU score fluctuates with more pretraining data), we revisit its definition. Recall that the common form of BLEU score is as follows

$$
\text{BLEU} = \text{brevity-penalty} \cdot \left( \prod_{i=1}^{4} \text{precision}_i \right)^{1/4}, \tag{4}
$$

where $\text{precision}_n$ refers to the precision of n-grams, and the second term is the geometric mean of the precision when n is varied from 1 to 4. In all the experiments, we observe brevity-penalty = 1, i.e., the non-monotonic behavior comes from the precision term, not the brevity penalty.

## 5.1 OTHER TASKS

In Appendix B, we show that the proposed log scaling law is not only applicable to the translation scores/tasks but also to metrics on question answering, reasoning, reading comprehension, and textual entailment tasks within SuperGLUE (Wang et al., 2019). Our results demonstrate that the same scaling law captures the scaling of these metrics as the pretraining data grows.

## 6 DISCUSSION AND CONCLUSION

We study the scaling behavior of the *downstream* performance in machine translation as the pretraining data grows and propose scaling laws for both *downstream* cross-entropy and translation quality metrics. We demonstrate through extensive experiments that the scaling behavior is significantly influenced by (1) the degree of alignment between the pretraining and the downstream data and (2) the finetuning dataset size. In favorable cases where the distributions are sufficiently aligned, we show that *downstream* translation quality, measured by translation scores, can be accurately predicted using a log scaling law. However, with less alignment, there are cases where translation scores fluctuate unpredictably whereas *downstream* cross-entropy improves monotonically. We also observe that when the finetuning dataset size is sufficiently large, pretraining has little to no value. Our findings highlight the importance of studying *downstream* performance metrics and not making decisions solely based on cross-entropy (whether upstream or downstream).

**Limitations.** Our work goes beyond cross-entropy loss to understand and predict the downstream model performance at scale. While the proposed laws fit the empirical data well and predict the translation scores at scale successfully when there is sufficient alignment, there are cases where these scores do not scale monotonically. Our work identifies many such cases; however, as mentioned in Remark 1, a more linguistic approach into alignment in translation could provide better understanding.

**Reproducibility Statement.** We used publicly available datasets and models, and specified their versions with proper citations in Section 4 and Appendix A. We provided details on the training procedure and hyperparameters for both pretraining and finetuning stages.

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

# A    ADDITIONAL EXPERIMENTAL DETAILS

For the T5-3B experiments, pretraining for 1M steps takes 15-20 hours and finetuning takes 5-7 hours on an 8x8 TPU. For the sake of anonymity, we are unable to provide further information on compute specifications at this time, but we will add details upon acceptance.

## A.1    MODEL ARCHITECTURES

We provide the architecture details of the T5-3B and T5-770M models in Tables 1 and 2. These models were initially introduced by Raffel et al. (2020).

Table 1: T5-3B (Raffel et al., 2020) architecture details.

| | |
|---|---|
| Embedding Dimension | 1024 |
| Number of Heads | 32 |
| Number of Encoder Layers | 24 |
| Number of Decoder Layers | 24 |
| Head Dimension | 128 |
| MLP Dimension | 16384 |

Table 2: T5-770M (Raffel et al., 2020) architecture details.

| | |
|---|---|
| Embedding Dimension | 1024 |
| Number of Heads | 16 |
| Number of Encoder Layers | 24 |
| Number of Decoder Layers | 24 |
| Head Dimension | 64 |
| MLP Dimension | 2816 |

## A.2    OPTIMIZING THE SCALING LAW COEFFICIENTS

In this section, we provide more details on how we optimize the coefficients of the scaling laws. Following Hoffmann et al. (2022), we use the Huber loss (Huber, 1992) to minimize overfitting to the outliers. Huber loss is particularly useful to suppress the effect of the outlier data points in the optimization problem. More specifically, if the data point with value $r$ is predicted by the law as $\hat{r}$, the loss for that data point would be

$$\ell_\delta(r, \hat{r}) = \begin{cases} \frac{1}{2}(r - \hat{r})^2 & \text{for } |r - \hat{r}| \leq \delta, \\ \delta \cdot (|r - \hat{r}| - \frac{1}{2}\delta) & \text{otherwise.} \end{cases} \tag{5}$$

Due to the numerical range difference between the COMET/BLEU score (between 0 and 100) and the *downstream* cross-entropy typically taking much smaller values, we use $\delta = 0.1$ for BLEU score law in (1) and $\delta = 1e - 3$ for the *downstream* cross-entropy law in (3).

For optimization, we use the L-BFGS algorithm (Nocedal, 1980). Specifically, for COMET/BLEU score law in (1), we solve

$$\min_{E, A, \alpha, \beta} \sum_{\text{Data point } i} \ell_\delta(\log f_i, \log \hat{f}(D_{pi})), \tag{6}$$

where $D_{pi}$ is the pretraining dataset size and $f_i$ is the COMET/BLEU score for the data point $i$, and $\hat{f}(\cdot)$ is the approximation for the optimal law $f(\cdot)$. Similarly, for the *downstream* cross-entropy loss law in (3), we solve

$$\min_{E,A,\alpha} \sum_{\text{Data point } i} \ell_\delta(\log L_i, \log \hat{L}(D_{pi})), \tag{7}$$

where $D_{pi}$ is the pretraining dataset size and $L_i$ is the *downstream* cross-entropy loss for the data point $i$, and $\hat{L}(\cdot)$ is the approximation for the optimal law $L(\cdot)$.

## B  SUPERGLUE EXPERIMENTS

Figure 6 demonstrates how SuperGLUE (Wang et al., 2019) task metrics such as Boolean Questions (BoolQ) (Clark et al., 2019), CommitmentBank (CB) (De Marneffe et al., 2019), Choice of Plausible Alternatives (COPA) (Roemmele et al., 2011), Multi-Sentence Reading Comprehension (MultiRC) (Khashabi et al., 2018), Recognizing Textual Entailment (RTE) (Dagan et al., 2006; Bar Haim et al., 2006; Giampiccolo et al., 2007; Bentivogli et al., 2009), Reading Comprehension with Commonsense Reasoning Dataset (ReCoRD) (Zhang et al., 2018), and Word-in-Context (WiC) (Pilehvar & Camacho-Collados, 2019) scale as the pretraining data grows. For these experiments, we use T5-3B model pretrained on en-MC4 data (same as Section 5). For finetuning on SuperGLUE, we use a batch size of 128 and a sequence length of 512 for 300 steps. We use a constant learning rate by selecting the best from $\{0.001, 0.005, 0.01, 0.05, 0.1, 0.5\}$.

The results indicate that the same scaling law, $\mathbf{f}(\mathbf{D_P}) = (\log(\mathbf{A} \cdot \mathbf{D_P^\alpha}))^\beta$, that was demonstrated to fit well to translation scores in Section 5 also captures the scaling of question answering (BoolQ, MultiRC), reasoning (COPA), reading comprehension (ReCoRD), and textual entailment (RTE) tasks, as well.

## C  ADDITIONAL EXPERIMENTAL RESULTS

In this section, we provide additional experimental results that we had to skip in the main body due to the page limit.

### C.1  RESULTS WITH COMET SCORES

We extend our experimental evaluation to COMET score, which we had to skip in the main body due to the page limit. In Figure 7, we provide the COMET scores for the models previously used in Figures 2 and 3 for BLEU score and cross-entropy. Similar to BLEU score, the law given in (1) well describes the scaling behavior of COMET score, when there is sufficient alignment between the pretraining and dowsntream data (Figure 7-(**top**)). When the alignment is not sufficient (Figure 7-(**bottom**)), again similar to the BLEU score, COMET score fluctuates and sometimes gets worse with more pretraining.

### C.2  RESULTS ON T5-770M

In Figures 8 and 9, we present results similar to Figures 2 and 3 in Section 5, but for T5-770M instead of T5-3B. In general, we observe a similar trend. The proposed scaling laws describe the downstream behavior well when the pretraining and downstream data are aligned. Similar to the results in T5-3B in the main body of the paper, in Figure 9-(*top, right*), we observe a break in the scaling law when the pretraining dataset is 100% en-MC4 and the task is en-fr translation – suggesting the same misalignment for this pretraining data and task that was also observed in Section 5 on the larger T5-3B model.

### C.3  OPTIMIZED COEFFICIENTS AND PREDICTION ERRORS OF THE SCALING LAWS

In Tables 3, 4, 5, and 6, we provide the optimized coefficients for the scaling laws plotted in Figures 2 and 3 together with the prediction error.

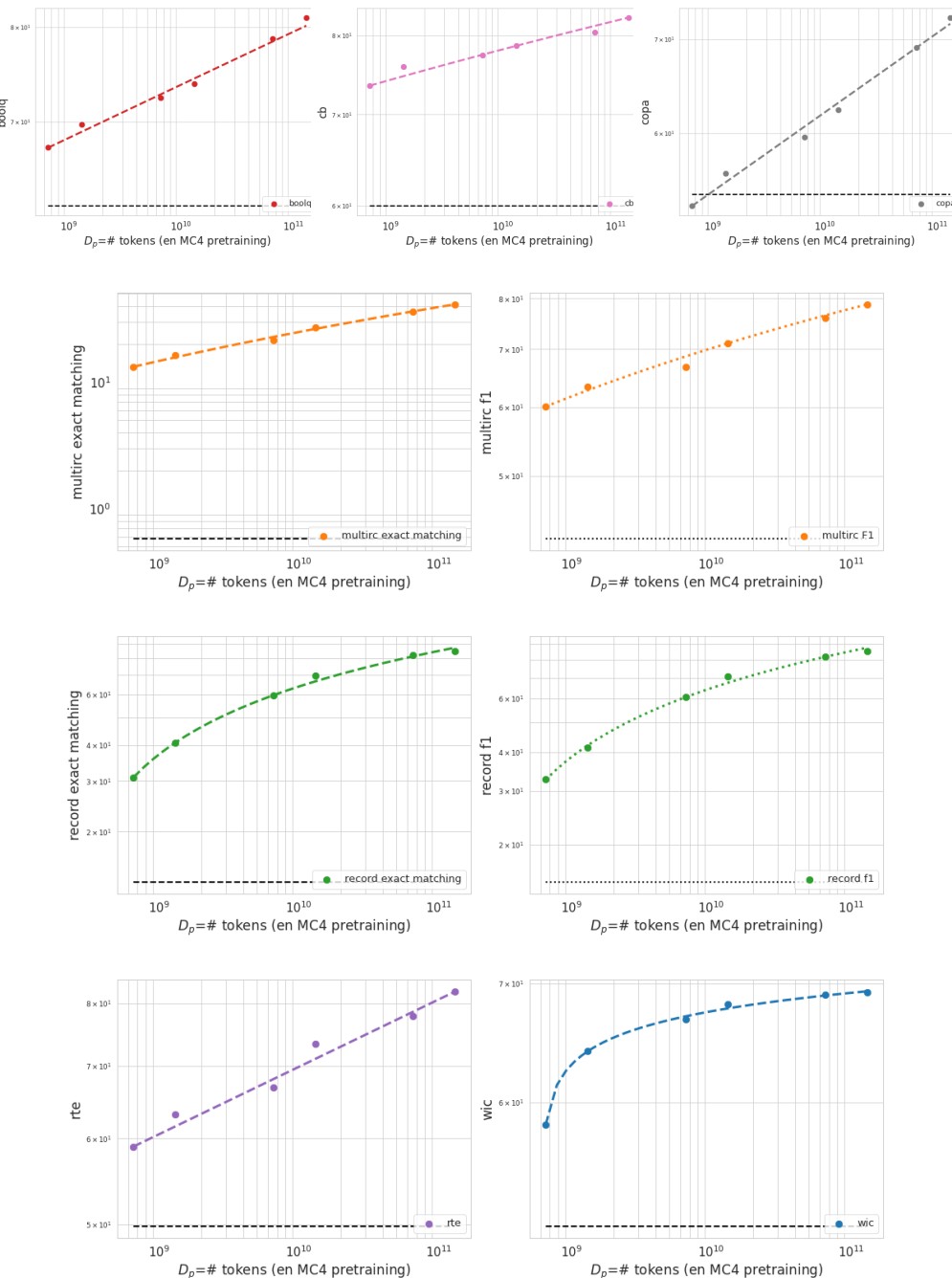

Figure 6: **SuperGLUE scores vs pretraining dataset size:** $\mathbf{f(D_p)} = (\log(\mathbf{A} \cdot \mathbf{D_p^{\alpha}}))^{\beta}$. Pretraining dataset is en-MC4 and finetuning dataset is SuperGLUE. For all the plots, the markers are the actual experimental results and the black horizontal curves correspond to the non-pretrained model directly trained on the SuperGLUE dataset.

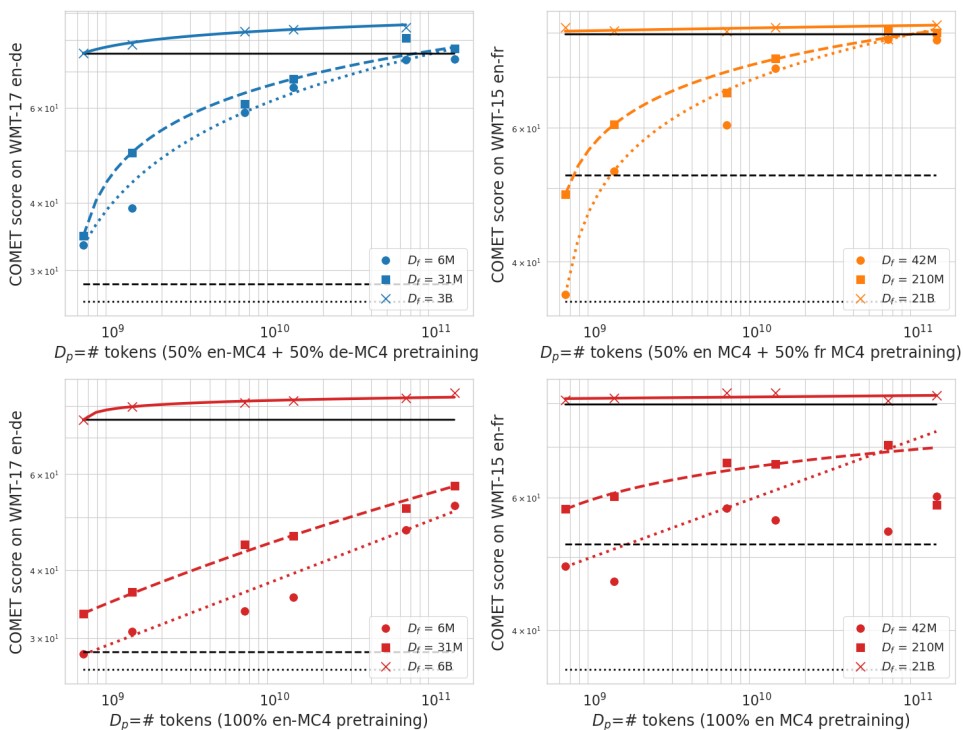

Figure 7: **(top) COMET score results for the** $50\%$-$50\%$ **balanced experiments in Figure 2:** $\mathbf{f}(\mathbf{D_p}) = (\log(\mathbf{A} \cdot \mathbf{D_p^{\alpha}}))^{\beta}$. **(left)** WMT-17 en-to-de translation task. Pretraining dataset has $50\%$ en-MC4 + $50\%$ de-MC4. Dotted, dashed, and solid blue curves correspond to the fitted scaling laws for different finetuning dataset sizes, $D_f = 6M$, $D_f = 31M$, $D_f = 3B$ tokens, respectively. **(right)** WMT-15 en-to-fr translation task. Pretraining dataset has $50\%$ en-MC4 and $50\%$ fr-MC4. Dotted, dashed, and solid orange curves correspond to the fitted scaling laws for different finetuning dataset sizes, $D_f = 42M$, $D_f = 210M$, $D_f = 21B$ tokens, respectively. **(bottom) COMET score results for the** $100\%$ **en-MC4 pretraining experiments in Figure 3:** Same as the top row, except that the pretraining dataset is $100\%$ en-MC4. For all the plots, the markers are the actual experimental results and the black horizontal curves correspond to the non-pretrained model directly trained on the task dataset. **The finetuning dataset size increases in the order of dotted-dashed-solid for all the curves including the black horizontal lines.**

Table 3: The coefficients for the BLEU score law $f(D_p) = (\log(A \cdot D_p^{\alpha}))^{\beta}$ for the results in Figure 2-**(top)**. For the BLEU score laws, we use $\delta = 0.1$ for the Huber Loss. We report $\log A$ instead of $A$ since $A$ typically takes very small and very large values.

| Pretraining Dataset | Finetuning Dataset | Finetuning Dataset Size | $\log A$ | $\alpha$ | $\beta$ | Prediction Error |
|---|---|---|---|---|---|---|
| $50\%$ en + $50\%$ de-MC4 | WMT-17 en-de | 6M | $-180.75$ | $9.00$ | $0.75$ | $0.034$ |
| $50\%$ en + $50\%$ de-MC4 | WMT-17 en-de | 31M | $-1.68 \times 10^3$ | $84.04$ | $0.49$ | $0.050$ |
| $50\%$ en + $50\%$ de-MC4 | WMT-17 en-de | 3B | $-1.64 \times 10^8$ | $9.91 \times 10^6$ | $0.19$ | $0.048$ |
| $50\%$ en + $50\%$ fr-MC4 | WMT-15 en-fr | 42M | $-1.82 \times 10^4$ | $8.98 \times 10^2$ | $0.42$ | $0.061$ |
| $50\%$ en + $50\%$ fr-MC4 | WMT-15 en-fr | 210M | $-2.33 \times 10^4$ | $1.21 \times 10^3$ | $0.40$ | $0.013$ |
| $50\%$ en + $50\%$ fr-MC4 | WMT-15 en-fr | 21B | $5.08 \times 10^3$ | $4.61 \times 10^8$ | $0.16$ | $0.005$ |
| $50\%$ en + $50\%$ ro-MC4 | WMT-16 en-ro | 625K | $-36.02$ | $1.77$ | $1.28$ | $0.042$ |
| $50\%$ en + $50\%$ ro-MC4 | WMT-16 en-ro | 3M | $-0.115.03$ | $5.69$ | $0.89$ | $0.015$ |
| $50\%$ en + $50\%$ ro-MC4 | WMT-16 en-ro | 312M | $-1.82 \times 10^4$ | $9.04 \times 10^2$ | $0.40$ | $0.015$ |

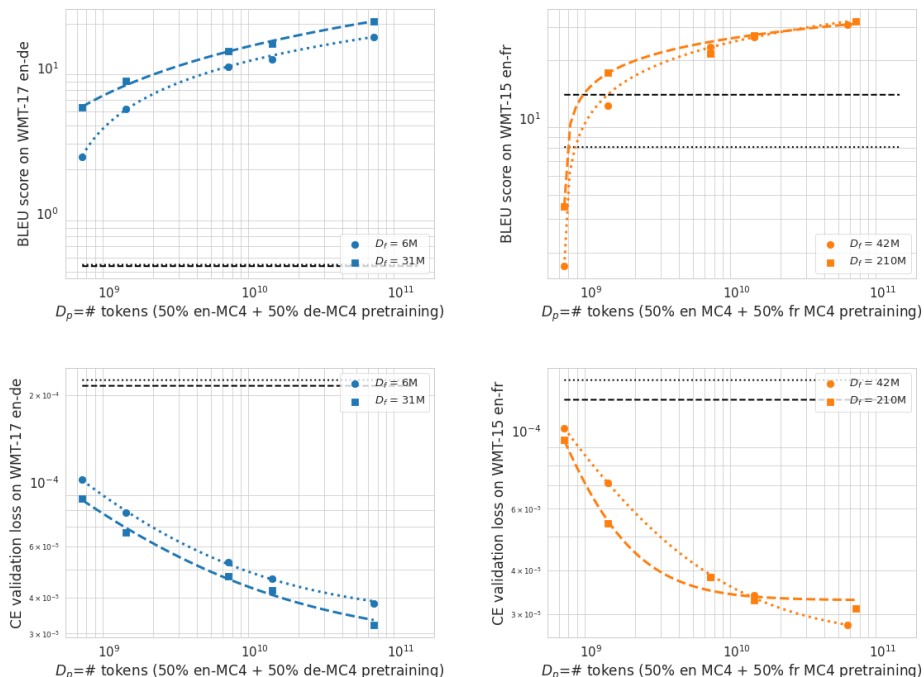

Figure 8: **(top) BLEU score vs pretraining dataset size:** $\mathbf{f(D_p)} = (\log(\mathbf{A} \cdot \mathbf{D_p^\alpha}))^\beta$. *(left)* WMT-17 en-to-de translation task. Pretraining dataset has $50\%$ en-MC4 + $50\%$ de-MC4. Dotted and dashed blue curves correspond to the fitted scaling laws for different finetuning dataset sizes, $D_f = 6M$ and $D_f = 31M$ tokens, respectively. *(right)* WMT-15 en-to-fr translation task. Pretraining dataset has $50\%$ en-MC4 and $50\%$ fr-MC4. Dotted and dashed orange curves correspond to the fitted scaling laws for different finetuning dataset sizes, $D_f = 42M$ and $D_f = 210M$ tokens, respectively. **(bottom) Cross-entropy (CE) validation loss vs pretraining dataset size:** $\mathbf{L(D_p)} = \mathbf{E} + \frac{\mathbf{A}}{\mathbf{D_p^\alpha}}$. Same models as the top row. For all the plots, the markers are the actual experimental results and the black horizontal curves correspond to the non-pretrained model directly trained on the task dataset. **The finetuning dataset size increases in the order of dotted-dashed for all the curves including the black horizontal lines.**

Table 4: The coefficients for the *downstream* cross-entropy law $L(D_p) = E + \frac{A}{D_p^\alpha}$ for the results in Figure 2-(**bottom**). For the *downstream* cross-entropy laws, we use $\delta = 10^{-5}$ for the Huber Loss.

| Pretraining Dataset | Finetuning Dataset | Finetuning Dataset Size | $E$ | $A$ | $\alpha$ | Prediction Error |
|---|---|---|---|---|---|---|
| $50\%$ en + $50\%$ de-MC4 | WMT-17 en-de | 6M | $3.21 \times 10^{-5}$ | $35.45$ | $0.64$ | $1.36 \times 10^{-12}$ |
| $50\%$ en + $50\%$ de-MC4 | WMT-17 en-de | 31M | $3.28 \times 10^{-5}$ | $4.70 \times 10^2$ | $0.78$ | $3.17 \times 10^{-12}$ |
| $50\%$ en + $50\%$ de-MC4 | WMT-17 en-de | 3B | $2.24 \times 10^{-5}$ | $2.56 \times 10^{-2}$ | $0.36$ | $5.76 \times 10^{-14}$ |
| $50\%$ en + $50\%$ fr-MC4 | WMT-15 en-fr | 42M | $2.72 \times 10^{-5}$ | $2.01 \times 10^6$ | $1.18$ | $7.52 \times 10^{-13}$ |
| $50\%$ en + $50\%$ fr-MC4 | WMT-15 en-fr | 210M | $2.57 \times 10^{-5}$ | $1.75 \times 10^7$ | $1.30$ | $2.24 \times 10^{-13}$ |
| $50\%$ en + $50\%$ fr-MC4 | WMT-15 en-fr | 21B | $1.11 \times 10^{-7}$ | $3.41 \times 10^{-5}$ | $1.82 \times 10^{-2}$ | $5.20 \times 10^{-14}$ |
| $50\%$ en + $50\%$ ro-MC4 | WMT-16 en-ro | 625K | $2.45 \times 10^{-5}$ | $0.49$ | $0.41$ | $3.61 \times 10^{-12}$ |
| $50\%$ en + $50\%$ ro-MC4 | WMT-16 en-ro | 3M | $2.62 \times 10^{-5}$ | $2.40$ | $0.49$ | $2.19 \times 10^{-12}$ |
| $50\%$ en + $50\%$ ro-MC4 | WMT-16 en-ro | 312M | $2.08 \times 10^{-5}$ | $3.94$ | $0.53$ | $5.95 \times 10^{-12}$ |

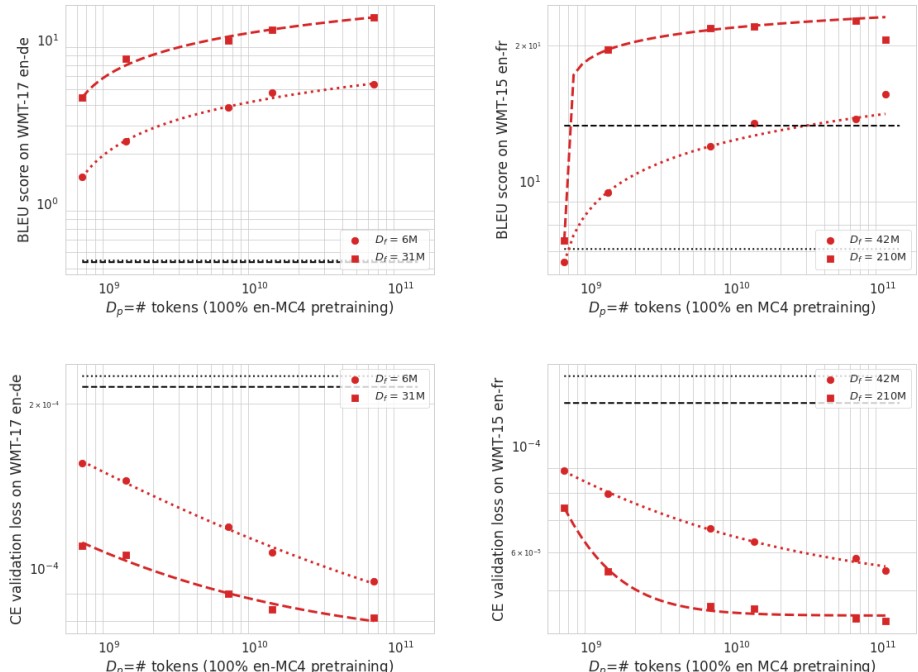

Figure 9: **(top) BLEU score vs pretraining dataset size:** $\mathbf{f}(\mathbf{D_p}) = (\log(\mathbf{A} \cdot \mathbf{D_p^\alpha}))^\beta$. *(left)* WMT-17 en-to-de translation task. Dotted and dashed red curves correspond to the fitted scaling laws for different finetuning dataset sizes, $D_f = 6M$ and $D_f = 31M$ tokens, respectively. *(right)* WMT-15 en-to-fr translation task. Dotted and dashed red curves correspond to the fitted scaling laws for different finetuning dataset sizes, $D_f = 42M$ and $D_f = 210M$ tokens, respectively. **(bottom) Cross-entropy (CE) validation loss vs pretraining dataset size:** $\mathbf{L}(\mathbf{D_p}) = \mathbf{E} + \frac{\mathbf{A}}{\mathbf{D_p^\alpha}}$. Same models as the top row. For all the plots, the markers are the actual experimental results and the black horizontal curves correspond to the non-pretrained model directly trained on the task dataset. **The finetuning dataset size increases in the order of dotted-dashed for all the curves including the black horizontal lines.**

Table 5: The coefficients for the BLEU score law $f(D_p) = (\log(A \cdot D_p^\alpha))^\beta$ for the results in Figure 3-(**top**). For the BLEU score laws, we use $\delta = 0.1$ for the Huber Loss. We report $\log A$ instead of $A$ since $A$ typically takes very small and very large values.

| Pretraining Dataset | Finetuning Dataset | Finetuning Dataset Size | $\log A$ | $\alpha$ | $\beta$ | Prediction Error |
|---|---|---|---|---|---|---|
| 100% en-MC4 | WMT-17 en-de | 6M | $-1.88$ | 0.15 | 3.30 | 0.014 |
| 100% en-MC4 | WMT-17 en-de | 31M | $-1.81 \times 10^4$ | 896.12 | 0.28 | 0.006 |
| 100% en-MC4 | WMT-17 en-de | 3B | $1.02 \times 10^{-7}$ | 104.92 | 0.42 | 0.015 |
| 100% en-MC4 | WMT-15 en-fr | 42M | 1.00 | $2.57 \times 10^{-5}$ | $1.11 \times 10^4$ | 0.042 |
| 100% en-MC4 | WMT-15 en-fr | 210M | $-6.38 \times 10^7$ | $3.43 \times 10^6$ | 0.20 | 0.034 |
| 100% en-MC4 | WMT-15 en-fr | 21B | 204.81 | $3.80 \times 10^{14}$ | $9.97 \times 10^{-3}$ | 0.004 |
| 100% en-MC4 | WMT-16 en-ro | 625K | $-10.54$ | 0.55 | 1.12 | 0.008 |
| 100% en-MC4 | WMT-16 en-ro | 3M | $-40.41$ | 2.11 | 0.79 | 0.025 |
| 100% en-MC4 | WMT-16 en-ro | 312M | 3.61 | $8.17 \times 10^5$ | 0.19 | 0.018 |

Table 6: The coefficients for the *downstream* cross-entropy law $L(D_p) = E + \frac{A}{D_p^\alpha}$ for the results in Figure 3-(**bottom**). For the *downstream* cross-entropy laws, we use $\delta = 10^{-5}$ for the Huber Loss.

| Pretraining Dataset | Finetuning Dataset | Finetuning Dataset Size | $E$ | $A$ | $\alpha$ | Prediction Error |
|---|---|---|---|---|---|---|
| 100% en-MC4 | WMT-17 en-de | 6M | $3.22 \times 10^{-13}$ | $3.18 \times 10^{-3}$ | 0.15 | $5.79 \times 10^{-12}$ |
| 100% en-MC4 | WMT-17 en-de | 31M | $3.24 \times 10^{-5}$ | $5.20 \times 10^{-3}$ | 0.20 | $9.25 \times 10^{-13}$ |
| 100% en-MC4 | WMT-17 en-de | 3B | $2.24 \times 10^{-5}$ | $2.56 \times 10^{-2}$ | 0.36 | $5.76 \times 10^{-14}$ |
| 100% en-MC4 | WMT-15 en-fr | 42M | $3.49 \times 10^{-5}$ | $1.05 \times 10^{-2}$ | 0.25 | $3.63 \times 10^{-13}$ |
| 100% en-MC4 | WMT-15 en-fr | 210M | $4.24 \times 10^{-5}$ | 19.39 | 0.66 | $5.40 \times 10^{-13}$ |
| 100% en-MC4 | WMT-15 en-fr | 21B | $1.26 \times 10^{-7}$ | $2.59 \times 10^{-5}$ | $4.81 \times 10^{-3}$ | $3.63 \times 10^{-14}$ |
| 100% en-MC4 | WMT-16 en-ro | 625K | $5.79 \times 10^{-12}$ | $1.03 \times 10^{-3}$ | $7.76 \times 10^{-2}$ | $5.56 \times 10^{-12}$ |
| 100% en-MC4 | WMT-16 en-ro | 3M | $1.78 \times 10^{-12}$ | $9.98 \times 10^{-4}$ | $8.33 \times 10^{-2}$ | $8.23 \times 10^{-12}$ |
| 100% en-MC4 | WMT-16 en-ro | 312M | $5.85 \times 10^{-5}$ | $1.37 \times 10^3$ | 0.88 | $3.05 \times 10^{-13}$ |

