# OpenReview forum: "Scaling Laws for Downstream Task Performance in Machine Translation"
_ICLR.cc/2025/Conference — ICLR 2025 Poster_

### Official Review · Reviewer_9RdU · 2024-10-29

**Soundness:** 4
**Presentation:** 3
**Contribution:** 3
**Rating:** 8
**Confidence:** 4

**Summary:**

This paper investigates scaling laws for downstream task performance in machine translation, specifically examining how pretraining dataset size affects translation quality after finetuning. The authors propose a novel log-law to describe the scaling behavior of translation quality metrics (BLEU, ROUGE, COMET) and demonstrate that the relationship between pretraining data size and downstream performance heavily depends on distribution alignment and finetuning dataset size. A key finding is that while cross-entropy consistently follows power-law scaling, translation quality metrics may deviate from expected scaling patterns when pretraining and downstream distributions are not well-aligned.

**Strengths:**

- Exceptionally well-grounded in literature, with relevant prior work effectively woven throughout the paper's narrative.
- Strong experimental methodology with fully controlled experiments that isolate different variables' effects. In contrast to a lot of recent LLM works, the experiments are fully controlled which I appreciate a lot. The paper demonstrates comprehensive validation across multiple metrics and scenarios, with clear empirical support for the proposed log-law.
- The writing is clear and the structure effective. Clear and systematic presentation of results with thorough analysis.
- The proposed log-law for translation quality metrics is well-motivated and comprehensively validated.
- Makes important observations challenging common assumptions about cross-entropy and translation quality metrics.

**Weaknesses:**

- Limited to pairwise language combinations in pretraining - would benefit from discussion about generalization to realistic multilingual pretraining scenarios.

- It would be a great addition to have a short discussion that connects these findings to recent successes in LLM-based MT. For instance, some works like https://aclanthology.org/2024.tacl-1.32/ and https://aclanthology.org/2024.acl-long.336/ fine-tuned directly on parallel data, but these LLMs are heavily English-centric and therefore there is a low distribution alignment. More recent work like ALMA https://openreview.net/forum?id=farT6XXntP and TowerLLM https://openreview.net/forum?id=EHPns3hVkj#discussion circumvent this issue by continuing pre-training on multilingual data before fine-tuning on parallel data.

- Minor  issues
    - Lines 142-143: "Tran et al., 2019" is repeated 3 times
    - Lines 264-265: it'd be nice to include fine-tuning dataset sizes here

**Questions:**

- This is in not a critique but I'm curious about your thoughts. You study pre-training with up to 2 languages. How do you think your findings translate to the more realistic setting of pre-training on a large number of languages, with severe misalignments in data size between the languages?

- Similarly, do you have any findings related to language similarity?

---

> ### Author Response · Authors · 2024-11-28
> **Thank you for your constructive feedback!**
>
> **1. Generalization to multiple languages:**
>
> In the revised paper, we provided a definition of alignment in Section 3.2 to be more concrete about what we meant by “more aligned” or “sufficiently aligned”:
>
> $$\mathcal{A}(D, T(L_{\text{source}}, L_{\text{dest}})) = P_{L_{\text{source}}} \cdot P_{L_{\text{dest}}} + 0.7 \cdot P_{L_{\text{source}}} + 0.8 \cdot P_{L_{\text{dest}}},$$
>
> where $ D$ is the pretraining data mixture, $T(L_{\text{source}}, L_{\text{dest}})$ is the translation task from $L_{\text{source}}$ to $L_{\text{dest}}$, $P_{L_{\text{source}}}$ is percentage of $L_{\text{source}}$ in $D$, and $P_{L_{\text{dest}}}$ is percentage of $L_{\text{dest}}$ in $D$.
>
> For instance, for an en-to-fr translation task, a pretraining data mixture with $50$% en and $50$% fr data would yield an alignment score of  $\mathcal{A}(D, T(\text{en}, \text{fr})) = 0.5 \cdot 0.5 + 0.7 \cdot 0.5 + 0.8 \cdot 0.5 = 1$. Likewise, a pretraining data mixture with $100$% en would have an alignment score of  $\mathcal{A}(D, T(\text{en}, \text{fr})) = 1 \cdot 0 + 0.7 \cdot 1 + 0.8 \cdot 0 = 0.7$.
>
>
>
> This definition can be used for multilingual pretraining data with more than two languages as well. In such a case, the alignment between the pretraining data D and a set of translation tasks $T_1$, …, $T_k$, can be measured as $ \frac{1}{K} \sum_{k=1}^K  \mathcal{A}(D,T_k)$.
>
> **2. Discussion on new references:**
>
> We thank the reviewer for bringing these references to our attention. Their findings indeed support our conclusions about alignment and we added a discussion on this in the related work section in the revised paper.
>
> **3. Language similarity:**
>
> While we can reason about the language similarity based on our findings, e.g., french-english might be less similar than german-english due to the trend in scaling laws, we think such a conclusion requires more linguistic analysis. However, we do believe that we managed to highlight some interesting cases that are worth attention linguistically.
>
> **4. Lines 142-143: "Tran et al., 2019" is repeated 3 times:**
>
> We thank the reviewer for catching this. We fixed this in the revised paper.
>
> **5. Lines 264-265: it'd be nice to include fine-tuning dataset sizes here:**
>
> We thank the reviewer for the suggestion. We included dataset sizes accordingly in the revised paper.
>
> -----
> We thank the reviewer for their time reading our paper and for providing very helpful and constructive feedback. If there is any remaining question or concern, we are happy to discuss further.

---

> > ### Comment · Reviewer_9RdU · 2024-11-28
> >
> > Thanks for your clarification! I’m keeping my score and I would like to see this paper accepted.

---

> > > ### Author Response · Authors · 2024-11-30
> > > **Thank you!**
> > >
> > > We thank the reviewer for their time reading our paper and rebuttal, and their constructive feedback. We appreciate their support!

---

### Official Review · Reviewer_nK1N · 2024-11-02

**Soundness:** 3
**Presentation:** 4
**Contribution:** 3
**Rating:** 8
**Confidence:** 3

**Summary:**

The paper investigates the scaling behavior of downstream performance metrics, including translation scores and cross-entropy loss, as a function of the size of the pretraining dataset in machine translation tasks. The authors conduct experiments using T5 models trained on different subsets of the Multilingual C4 dataset and fine-tuned on various translation tasks. They find that when the pretraining and downstream tasks have well-aligned distributions, both translation scores and cross-entropy loss improve monotonically with more pretraining data. However, when the distributions are not well-aligned, translation scores can exhibit non-monotonic behavior, while cross-entropy loss still improves monotonically. The authors propose a practical guide for evaluating the value of pretraining data for translation tasks based on downstream translation scores and demonstrate the effectiveness of their approach on multiple natural language processing tasks.

**Strengths:**

The paper's main contribution is its in-depth investigation of the impact of pre-training data on machine translation tasks and the proposal of a new logarithmic law to describe the change in downstream performance as the size of the pre-training data increases. This research is helpful to understand the performance of large-scale language models in specific downstream tasks and provides a new method for evaluating the value of pre-training data. From the perspectives of innovation and importance, the paper is highly valuable. Additionally, the quality and clarity of the paper are good, with detailed descriptions of experimental design and result analysis, and abundant charts and empirical evidence to support conclusions.

**Weaknesses:**

One potential weakness of the paper is it is focusing solely on downstream performance changes in machine translation tasks. This limits the comprehensive evaluation of the applicability of the logarithmic law to a wider range of natural language processing tasks.
For NLP tasks that use precision and recall as metrics, can these be effectively aligned with the pre-training procedure? Additionally, can cross-entropy be generalized to observe the scaling law?

It is recommended to include a reference to the inference scaling law from OpenAI’s o1 model and to discuss the scaling law within a broader framework.

**Questions:**

The alignment between the pre-training and downstream task is crucial to understanding the scaling law. I am not entirely clear on what “align” means in this context. Could you please explain in detail the meaning of “align” in this work?

**Details Of Ethics Concerns:**

No.

---

> ### Author Response · Authors · 2024-11-27
> **Thank you for your constructive feedback!**
>
> **1. Applicability to other NLP tasks:**
>
> We agree that it is important to see to what extent the proposed logarithmic law applies to other NLP tasks. We actually provided empirical evidence in Figure 6 of Appendix that the same law successfully explains the scaling behavior of SuperGLUE benchmark scores, including Boolean Ques-tions (BoolQ), CommitmentBank (CB), Choice ofPlausible Alternatives (COPA), Multi-Sentence Reading Comprehension (MultiRC), Recognizing Textual Entailment (RTE), Reading Comprehension with Commonsense Reasoning Dataset (ReCoRD), and Word-in-Context(WiC).
>
> **2. Can cross entropy be generalized to observe the scaling law?**
>
> Almost all existing scaling laws were proposed for cross entropy loss. We are unsure if we understand the reviewer’s question correctly and we are happy to discuss further if they can clarify their question a bit more.
>
> **3. Reference to OpenAI’s o1 model:**
>
> We thank the reviewer for this suggestion. The inference scaling law in OpenAI’s o1 model relates the train time and test time of the reinforcement learning step with the AIME accuracy. Our setup is different since we relate pretraining data to downstream performance with a finetuning step in between. We added the relevant reference in the revised paper.
>
> **4. Definition of alignment:**
>
> We use the alignment concept as a measure of how much language overlap there is between pretraining data and translation task, with a slightly higher weight on the target language rather than the source language. In the revised paper, we added a definition of alignment that measures language similarity between the pretraining and downstream task. Please see Definition 1 in Section 3.2 for the translation alignment score that reflects the way we interpret alignment for translation tasks:
>
> $$\mathcal{A}(D, T(L_{\text{source}}, L_{\text{dest}})) = P_{L_{\text{source}}} \cdot P_{L_{\text{dest}}} + 0.7 \cdot P_{L_{\text{source}}} + 0.8 \cdot P_{L_{\text{dest}}},$$
>
> where $ D$ is the pretraining data mixture, $T(L_{\text{source}}, L_{\text{dest}})$ is the translation task from $L_{\text{source}}$ to $L_{\text{dest}}$, $P_{L_{\text{source}}}$ is percentage of $L_{\text{source}}$ in $D$, and $P_{L_{\text{dest}}}$ is percentage of $L_{\text{dest}}$ in $D$.
>
> For instance, for an en-to-fr translation task, a pretraining data mixture with $50$% en and $50$% fr data would yield an alignment score of  $\mathcal{A}(D, T(\text{en}, \text{fr})) = 0.5 \cdot 0.5 + 0.7 \cdot 0.5 + 0.8 \cdot 0.5 = 1$. Likewise, a pretraining data mixture with $100$% en would have an alignment score of  $\mathcal{A}(D, T(\text{en}, \text{fr})) = 1 \cdot 0 + 0.7 \cdot 1 + 0.8 \cdot 0 = 0.7$.
>
> We also included the corresponding scores in Figures 2, 3, 4, and 5 for each experiment.
>
> -----
> We thank the reviewer for their time reading our paper and for providing very helpful and constructive feedback. If there is any remaining question or concern, we are happy to discuss further.

---

> > ### Author Response · Authors · 2024-12-02
> > **Any remaining questions/concerns?**
> >
> > We appreciate the reviewer's time and constructive feedback. We tried to address their points in the rebuttal and revised our paper accordingly. If there are any remaining questions or concerns, we are happy to discuss before the discussion period ends.

---

### Official Review · Reviewer_f7Dz · 2024-11-05

**Soundness:** 3
**Presentation:** 3
**Contribution:** 2
**Rating:** 8
**Confidence:** 3

**Summary:**

The paper analyses to what extent scaling laws apply if the pre-training and downstream data are well or less well aligned. They find that iff the data is less aligned, task-specific metrics can fluctuate when increasing the pre-training data, while increasing more monotonically if the data is well aligned. Their findings are mostly based on machine translation experiments but also generalize to more general tasks such as the tasks in the SuperGlue benchmark set.

**Strengths:**

- well written paper
- addresses a problem that is relevant for a large community
- thourough empirical analysis

**Weaknesses:**

Although the concept of alignement is central to this work and many times the authors refer to "degree of" or "not sufficient" alignment, no definition of alignment is given. I understand that this is not trivial but I would appreciate at least an attempt to provide a---ideally formal---definition that could also be used to quantify alignment. Currently, it seems the only way is to determine the degree in hindsight (see lines 227--228) by failing to fit the scaling law.

**Questions:**

To what extent do you expect your findings to carry over to decoder-only models more commonly found in current LLMs?

---

> ### Author Response · Authors · 2024-11-27
> **Thank you for your constructive feedback!**
>
> **1. Definition of Alignment:**
>
> We use the alignment concept as a measure of how much language overlap there is between pretraining data and translation task, with a slightly higher weight on the target language rather than the source language. In the revised paper, we added a definition of alignment that measures language similarity between the pretraining and downstream task. Please see Definition 1 in Section 3.2 for the translation alignment score that reflects the way we interpret alignment for translation tasks:
>
> $$\mathcal{A}(D, T(L_{\text{source}}, L_{\text{dest}})) = P_{L_{\text{source}}} \cdot P_{L_{\text{dest}}} + 0.7 \cdot P_{L_{\text{source}}} + 0.8 \cdot P_{L_{\text{dest}}},$$
>
> where $ D$ is the pretraining data mixture, $T(L_{\text{source}}, L_{\text{dest}})$ is the translation task from $L_{\text{source}}$ to $L_{\text{dest}}$, $P_{L_{\text{source}}}$ is percentage of $L_{\text{source}}$ in $D$, and $P_{L_{\text{dest}}}$ is percentage of $L_{\text{dest}}$ in $D$.
>
> For instance, for an en-to-fr translation task, a pretraining data mixture with $50$% en and $50$% fr data would yield an alignment score of  $\mathcal{A}(D, T(\text{en}, \text{fr})) = 0.5 \cdot 0.5 + 0.7 \cdot 0.5 + 0.8 \cdot 0.5 = 1$. Likewise, a pretraining data mixture with $100$% en would have an alignment score of  $\mathcal{A}(D, T(\text{en}, \text{fr})) = 1 \cdot 0 + 0.7 \cdot 1 + 0.8 \cdot 0 = 0.7$.
>
> We also included the corresponding scores in Figures 2, 3, 4, and 5 for each experiment.
>
> **2. Extensions:**
>
> Our focus in the paper is to understand the effect of the pretraining data and its size on the downstream performance. While we did not study how the model architecture or model size would affect the downstream performance, we do not expect this to change our conclusions about the effect of data and its size.
>
> ------
> We thank the reviewer for their time reading our paper and for providing very helpful and constructive feedback. If there is any remaining question or concern, we are happy to discuss further.

---

> > ### Author Response · Authors · 2024-12-02
> > **Any remaining questions/concerns?**
> >
> > We appreciate the reviewer's time and constructive feedback. We tried to address their points in the rebuttal and revised our paper accordingly. If there are any remaining questions or concerns, we are happy to discuss before the discussion period ends.

---

### Official Review · Reviewer_tpup · 2024-11-06

**Soundness:** 3
**Presentation:** 2
**Contribution:** 3
**Rating:** 6
**Confidence:** 4

**Summary:**

Past work on scaling laws has mostly focused on pretraining loss rather than downstream transfer learning task performance. This paper attempts to establish scaling laws for transfer learning for machine translation. The paper specifically investigates how pretraining data size affects downstream translation performance, measured in terms of downstream cross-entropy loss as well as in terms of downstream machine translation metrics like BLEU. Results show that that size of finetuning data, as well as the similarity between the pretraining and finetuning datasets, have a large impact on downstream results and scaling behavior. Most interestingly, the results indicate that when pretrain and finetune are well enough aligned, there is a clear log scaling law for pretraining size on downstream machine translation metric performance. However, when pretrain and finetune are dissimilar, more pretraining data can sometimes even hurt performance on downstream machine translation metrics. This stands in contrast with downstream cross-entropy, which improves monotonically with pretrain size regardless of pretrain/finetune mismatch.

**Strengths:**

The paper seeks empirical answers to important questions -- scaling laws (even rough ones) for downstream transfer can provide useful guidance: When should one seek out different pretraining data vs more pretraining data? How much finetuning data is sufficient for good downstream performance?

The paper focuses specifically on machine translation -- an important NLP task that has not been the main focus of prior transfer learning scaling law work. This is a strength. The application focus lets the paper go into more depth in empirical analysis and leads to clearer takeaways.

The paper concretely shows that downstream cross-entropy can be misleading. The paper finds that downstream cross-entropy monotonically improves with pretrain size, even when other machine translation metrics like BLEU do not. This is a useful result for future researchers.

**Weaknesses:**

Aspects of the presentation could be improved. Specifically, as a reader, I found it confusing that the paper essentially starts with a results / takeaways section before discussing experimental details. Then an experimental details section comes next, followed by a results / takeaways section. I think a more traditional ordering could be more effective here -- the intro could be used to discuss high-level takeaways up front, leaving the rest of the detailed results to be described after experimental setup is clear. Without having a clear understanding of experimental setup, I was left wondering in many cases how to interpret the early results discussion.

The one experimental weakness that stood out to me: It's not entirely clear how "alignment" between pretrain and finetune is being defined. For the most part, it seems to mean in the context of this paper whether and to what extend that datasets share languages. This could be further clarified and formalized -- but, further, more nuanced measures of alignment could easily be reported. These might help clarify experimental takeaways having to do with alignment, which represent some of the more interesting results in this paper.

Minor:

-Discussion of results on other tasks is odd given that paper is focused on MT.

-I couldn't follow remark 2 -- what was the main takeaway / hypothesis? Consider rephrasing this section for improved clarity.

-"3.4 A Guide for ..." section is kind of confusing -- especially part 2.

**Questions:**

see above

---

> ### Author Response · Authors · 2024-11-27
> **Thank you for your constructive feedback!**
>
> **1. Presentation:**
>
> We thank the reviewer for the feedback about the organization of the paper. We revised the paper, following their suggestions. We moved the highlight of the main results (Figure 1) from the introduction section to the methods section and connected it to the new section on alignment definition (Section 3.2).
>
> **2. Definition of alignment:**
>
> The reviewer asks valuable questions about the definition of “alignment” and how we can interpret this in the context of this paper. As the reviewer correctly stated, we use the alignment concept as a measure of how much language overlap there is between pretraining data and translation task, with a slightly higher weight on the target language rather than the source language. In the revised paper, we added a definition of alignment that measures language similarity between the pretraining and downstream task. Please see Definition 1 in Section 3.2 for the translation alignment score that reflects the way we interpret alignment for translation tasks:
>
> $$\mathcal{A}(D, T(L_{\text{source}}, L_{\text{dest}})) = P_{L_{\text{source}}} \cdot P_{L_{\text{dest}}} + 0.7 \cdot P_{L_{\text{source}}} + 0.8 \cdot P_{L_{\text{dest}}},$$
>
> where $ D$ is the pretraining data mixture, $T(L_{\text{source}}, L_{\text{dest}})$ is the translation task from $L_{\text{source}}$ to $L_{\text{dest}}$, $P_{L_{\text{source}}}$ is percentage of $L_{\text{source}}$ in $D$, and $P_{L_{\text{dest}}}$ is percentage of $L_{\text{dest}}$ in $D$.
>
> For instance, for an en-to-fr translation task, a pretraining data mixture with $50$% en and $50$% fr data would yield an alignment score of  $\mathcal{A}(D, T(\text{en}, \text{fr})) = 0.5 \cdot 0.5 + 0.7 \cdot 0.5 + 0.8 \cdot 0.5 = 1$. Likewise, a pretraining data mixture with $100$% en would have an alignment score of  $\mathcal{A}(D, T(\text{en}, \text{fr})) = 1 \cdot 0 + 0.7 \cdot 1 + 0.8 \cdot 0 = 0.7$.
>
> We also included the corresponding scores in Figures 2, 3, 4, and 5 for each experiment.
>
> **3. Remark 2:**
>
> We thank the reviewer for pointing out the unclarity in Remark 2. In remark 2, we wanted to understand how each of the terms in BLEU score behaves as the pretraining data increases, to better understand which term(s) is responsible for the non-monotonic behavior of the BLEU score especially when the alignment is not sufficient, i.e., when the BLEU score fluctuates with more pretraining data. We found out that the brevity-penalty is always 1 in our experiments. This means, the non-monotonic behavior comes from the precision term. We clarified these take-aways in the revised paper.
>
> **4. Guide for Data Valuation:**
> We revised the second step of the data valuation guide. We thank the reviewer for pointing out the unclarity.
>
> -----
> We thank the reviewer for their time reading our paper and for providing very helpful and constructive feedback. If there is any remaining question or concern, we are happy to discuss further.

---

> > ### Author Response · Authors · 2024-12-02
> > **Any remaining questions/concerns?**
> >
> > We appreciate the reviewer's time and constructive feedback. We tried to address their points in the rebuttal and revised our paper accordingly. If there are any remaining questions or concerns, we are happy to discuss before the discussion period ends.

---

### Official Review · Reviewer_2Keq · 2024-11-07

**Soundness:** 2
**Presentation:** 1
**Contribution:** 2
**Rating:** 3
**Confidence:** 4

**Summary:**

The paper proposes scaling laws for pre-training data and downstream data in the context of machine translation. The authors demonstrate that when the pre-training data is sufficiently aligned with the downstream task, both the downstream cross-entropy (CE) loss and translation quality adhere to the proposed scaling laws. However, if the pre-training data is not well-aligned, the downstream CE may still align with the scaling laws, but other metrics of translation quality may exhibit misalignment.

**Strengths:**

The paper conducts extensive experiments and present some interesting findings.

**Weaknesses:**

1. "Inaccurate" statement: The field of machine translation is increasingly adopting decoder-only architectures, whereas this paper focuses exclusively on encoder-decoder models. This focus leads to findings that conflict with those from SoTA decoder-only translation models. For example, in lines 355-359, the authors claim that "there is no need to pretrain the models when the fine-tuning data is large enough." This assertion totally contradicts the findings in [1], which states that "you only need a large amount of data for pre-training and a small number of data for continued fine-tuning." The key difference is that [1] employs a decoder-only model but it achieves top performance among other SoTA translation models. I am inclined to agree with the latter explanation because your training and test data are from the same domain (WMT). Training on WMT data usually yields better results on the WMT test set (especially BLEU), which might give the illusion that using only fine-tuning data is sufficient. However, this approach may lead to poor generalization abilities. A large training dataset may have less impact on in-domain test data, but it's essential to ensure that the model generalizes well to out-of-domain data. Therefore, the authors should also investigate the model's performance on other out-of-domain test datasets to provide a more comprehensive evaluation before make this statement.

2. Out-of-date metric: Most of the findings in this paper rely on BLEU scores. However, BLEU is increasingly regarded as an inaccurate metric that does not align well with human judgments, especially when compared to other metrics like COMET. The inclusion of ROUGE also seems weird, as ROUGE is primarily used for summarization evaluation rather than translation. While it might be applicable in certain cases, it is not a mainstream choice for assessing translation quality. The authors should consider using more suitable metrics, such as BLEURT, to provide a more accurate evaluation of their models.

3. Unclear writing: While this is a minor issue, the paper's writing feels disjointed and contains redundant information. There are several instances where the same points are reiterated, such as "when the distribution is well-aligned, both metrics and CE align with the scaling laws." Moreover, the frequent references to figures that are located several pages away disrupt the reading flow and make it difficult to follow the arguments being presented.

4. Lack of definition: I struggled to understand what the authors mean by the "distribution of pre-training and downstream task." It wasn't until midway through the paper that I realized this likely refers to the amount of monolingual target language data included in the pre-training dataset—where more target language data equates to a more "aligned distribution." One question here, if the authors were to conduct experiments on another low-resource language like Icelandic, where the fine-tuning data consists of a small amount of news content but the available pre-training monolingual data is primarily from games, the "distribution/alignment" becomes ambiguous. Alternatively, one might employ cross-lingual learning using a high-resource language like German, for which abundant news data is available. In this scenario, which dataset constitutes a "more aligned distribution"—the Icelandic data from a different domain or the German data from the same domain? I believe that the term "distribution" is too abstract and makes the paper challenging to understand. Providing a more precise definition or elaborating on this concept would greatly enhance the clarity of the paper.

5. A narrow scope in high-resource languages: The focus on high-resource languages is not particularly compelling, as SoTA models already perform exceptionally well in these languages, regardless of scaling laws. People can simply amass large amounts of data for training to achieve excellent results. The paper's emphasis on utilizing billions of tokens for pre-training and millions of parallel sentences for fine-tuning is not practical for many mid- or low-resource languages, where those languages do not have the resource.  A more interesting and valuable direction would be to explore scaling laws that examine how high-resource languages can be used to support low-resource languages. Investigating how data from other languages can enhance translation quality in languages with limited resources would address a critical need in the field of MT and contribute to more inclusive and effective translation models.

Update: I just found that I forgot adding the references:

References:

[1] A Paradigm Shift in Machine Translation: Boosting Translation Performance of Large Language Models

**Questions:**

N/A

---

> ### Author Response · Authors · 2024-11-27
> **Thank you for your constructive feedback!**
>
> **1. Clarification on value of pretraining vs finetuning data size”:**
>
> We understand that the conclusion from our empirical observations was not communicated clearly enough. We empirically observed that, in the particular setup we considered, as the finetuning data size grows, pretraining brings less value. This is an empirical finding, which led to the conclusion that “the value of pretraining data depends on the size of the finetuning data”. Notice that for the specific setup we considered, it is indeed **empirically evident** that pretraining does not bring almost any value when the finetuning data is large enough. However, **we never state this as a general suggestion** since high-quality finetuning data is more expensive and harder to obtain than unsupervised pretraining data. So, in practice, we typically do not have sufficiently large finetuning data and we agree that pretraining is still recommended before finetuning the model on supervised data, which aligns with the common practice, the reviewer’s comments, and other papers’ findings, e.g.  the ones stating that “we need large pretraining and a small finetuning data”. We apologize for this confusion, we revised our paper to clarify this point further, see the text in blue in pages 7-8.
>
> **2. Metrics:**
>
> The reviewer stated that BLEU score is not as aligned with human judgments as other metrics like COMET. In case this was missed during their reading, we actually provided results with COMET score as well. In particular, Figure 1, Section C.1, and Figure 7 show the scaling behavior of COMET score under the setups considered in the paper. We found that the COMET results are similar to BLEU results, in that the COMET score is also predictable with the proposed scaling law as long as there is sufficient alignment.
>
> **3. Writing and Organization:**
>
> We thank the reviewer for their constructive comments. We reorganized the paper and moved figures closer to where they are referenced.
>
> **4. Alignment Definition:**
>
> The reviewer asks valuable questions about the definition of “alignment” and how we can interpret this in the context of this paper. As the reviewer correctly stated, we use the alignment concept as a measure of how much language overlap there is between pretraining data and translation tasks, with a slightly higher weight on the target language rather than the source language. A more general definition of alignment, e.g. for the games vs news, would be an important contribution to the field and definitely deserves attention from the community. In the revised paper, we added a definition of alignment that measures language similarity between the pretraining and downstream task. Please see Definition 1 in Section 3.2 for the translation alignment score that reflects the way we interpret alignment for translation tasks:
>
> $$\mathcal{A}(D, T(L_{\text{source}}, L_{\text{dest}})) = P_{L_{\text{source}}} \cdot P_{L_{\text{dest}}} + 0.7 \cdot P_{L_{\text{source}}} + 0.8 \cdot P_{L_{\text{dest}}},$$
>
> where $ D$ is the pretraining data mixture, $T(L_{\text{source}}, L_{\text{dest}})$ is the translation task from $L_{\text{source}}$ to $L_{\text{dest}}$, $P_{L_{\text{source}}}$ is percentage of $L_{\text{source}}$ in $D$, and $P_{L_{\text{dest}}}$ is percentage of $L_{\text{dest}}$ in $D$.
>
> For instance, for an en-to-fr translation task, a pretraining data mixture with $50$% en and $50$% fr data would yield an alignment score of  $\mathcal{A}(D, T(\text{en}, \text{fr})) = 0.5 \cdot 0.5 + 0.7 \cdot 0.5 + 0.8 \cdot 0.5 = 1$. Likewise, a pretraining data mixture with $100$% en would have an alignment score of  $\mathcal{A}(D, T(\text{en}, \text{fr})) = 1 \cdot 0 + 0.7 \cdot 1 + 0.8 \cdot 0 = 0.7$.
>
> We also included the corresponding scores in Figures 2, 3, 4, and 5 for each experiment.
>
> **5. Suggestion for future work:**
>
> This is a very interesting and compelling idea. We believe this would be a great direction for future work and we think our work serves as an initial step towards understanding scaling behavior of translation quality.
>
> -------------
> We thank the reviewer for their time reading our paper and for providing very helpful and constructive feedback. If there is any remaining question or concern, we are happy to discuss further.

---

> > ### Comment · Reviewer_2Keq · 2024-11-30
> >
> > 1. As the authors claimed, their findings and conclusions are scenario-specific, limited to particular cases such as the encoder-decoder architecture and the WMT dataset, making them hard to generalize to other cases. While this is an interesting finding, the main issue lies in the lack of generalizability, as the findings are overly specific.
> >
> > 2. I did not overlook COMET in the paper; however, the other two metrics are inappropriate. While it is good that COMET is included, two out of the three metrics in the paper are, in my view, unsuitable.
> >
> > 3. Thanks for the authors for further stating the definition of the alignment. This is clearer now.
> >
> > I appreciate the authors' response, as well as their extensive experiments and interesting findings. However, the paper has two major issues. First, the findings are highly scenario-specific. They vary significantly when there are changes in the architecture, dataset, or languages involved. highly limiting generalizability. Second, the findings contribute little to advancing machine translation. The translation performance for the high-resource languages studied reported in the paper is already saturated. Even without the insights provided by the paper, practitioners would likely follow the same approach: fine-tuning a pre-trained encoder-decoder model with all available datasets, which is also a very standard way to do. Moreover, encoder-decoder models studied in the paper are gradually being replaced by alternative architectures for translation tasks. This makes the findings less relevant and, frankly, somewhat trivial for the field. A more valuable direction would have been to explore how scaling laws affect the amount of high-resource data required to improve low-resource translation performance (though this is beyond the scope of the current paper).
> >
> > Overall, I do not see a significant contribution in this work and therefore stand by my initial score.

---

> > > ### Author Response · Authors · 2024-11-30
> > > **Clarifications**
> > >
> > > We thank the reviewer for reading our rebuttal in detail. We would like to clarify three things:
> > >
> > > ----
> > >
> > > 1. First, we would like to clarify that the goal of the paper is not to improve machine translation, but rather, to point out discrepancies between different metrics. This is important since almost all existing scaling laws predict cross entropy at scale, which, as we show in the paper, does not always align with the downstream benchmark metrics. We believe **our results are alarming for practitioners who used to make their data selection choices based on cross entropy scaling laws since our paper shows that this could be misleading in some cases**. The impact we hope to make is not to improve machine translation but to show the broader language modeling community that we should pay more attention to scaling behavior of downstream metrics, which is not very well understood as opposed to the scaling behavior of cross entropy. We believe our paper serves the purpose of understanding the challenges in this space. We will revise the introduction section to clarify our goal and our message to the community.
> > >
> > > ------
> > >
> > > 2.  **Even without the insights provided by the paper, practitioners would likely follow the same approach: fine-tuning a pre-trained encoder-decoder model with all available datasets, which is also a very standard way to do.**:
> > >
> > > The reviewer is right and this is definitely not the contribution we wanted to convey in the paper. Our contribution here is not to say that "you should finetune on all the available data" since this is the most natural thing to do as the reviewer says. What we systematically show in our paper is that the value of pretraining decreases as we have more finetuning data. This is important because it suggests that it is essential to take the size of the finetuning data into account when we valuate different pretraining data, e.g. when we want to estimate market value of data, because the value of pretraining is different for different finetuning data sizes. For instance, if we have very limited finetuning data for a task, then a relevant pretraining data might be extremely valuable and we can estimate this value with the proposed scaling laws. However, if the conditions change and if we have more finetuning data, then the value of pretraining data reduces accordingly and it needs to be estimated with the correct finetuning data size. We hope the message we wanted to convey based on the observation on finetuning data size vs value of pretraining is more clear now. We will make sure to clarify this in the next revision.
> > >
> > > --------
> > >
> > > 3. **I did not overlook COMET in the paper; however, the other two metrics are inappropriate:**
> > >
> > > The reviewer mentioned in their first review that we should have studied metrics like COMET score and then we pointed out that we actually have results on COMET score. We are not sure why including BLEU and ROUGE scores, two other common translation scores, in addition to COMET, is "inappropriate". If we missed any ethical concerns regarding these scores, we are happy to revise our paper and remove these scores. But as far as we know, all three scores are widely used in machine translation papers published in top machine learning conferences. Below, we list a couple them that use BLEU and/or ROUGE scores to evaluate translation tasks:
> > >
> > >
> > > - (BLEU) [Order Matters in the Presence of Dataset Imbalance for Multilingual Learning](https://proceedings.neurips.cc/paper_files/paper/2023/file/d346609ec2fefd3938c898a0dda4a480-Paper-Conference.pdf), **NeurIPS** 2023.
> > >
> > > - (BLEU) [Data scaling laws in NMT: The effect of noise and architecture](https://proceedings.mlr.press/v162/bansal22b/bansal22b.pdf) **ICML**, 2022.
> > >
> > > - (BLEU) [Examining scaling and transfer of language model architectures for machine translation](https://proceedings.mlr.press/v162/zhang22h/zhang22h.pdf) **ICML**, 2022.
> > >
> > > - (BLEU) **ICLR 2022 Spotlight**: [Scaling laws for neural machine translation](https://openreview.net/pdf/dec3d7582a0893c49661157564fdbe66ccc0036f.pdf), **ICLR**, 2022.
> > >
> > > - (BLEU and ROUGE) [Expert-level protocol translation for self-driving labs](https://openreview.net/pdf?id=qXidsICaja), **NeurIPS**, 2024.
> > >
> > > - (BLEU) [BranchNorm: Robustly Scaling Extremely Deep Transformers](https://aclanthology.org/2024.findings-acl.695.pdf), **ACL** 2024.
> > >
> > > - (BLEU) [Multilingual Machine Translation with Large Language Models: Empirical Results and Analysis](https://aclanthology.org/2024.findings-naacl.176.pdf), **NAACL** 2024.
> > >
> > >
> > >
> > > Again, if there are any ethical concerns about these scores that we are not aware of, we would like to take action and remove the evaluations of these scores. Otherwise, we hope the results on COMET score address the reviewer's initial point about including COMET score in the evaluations.
> > >
> > > -----
> > >
> > >  We thank the reviewer for engaging in a discussion with us. If there are any remaining concerns, we are happy to discuss further.

---

> ### Comment · Reviewer_2Keq · 2024-12-01
>
> Thank you to the authors for their additional response; I greatly appreciate the effort. Regarding points 1 and 2, I acknowledge that the study's goals and findings provide valuable insights, alarming specific considerations for researchers and practitioners in machine translation. However, as I previously mentioned, the findings appear to be overly scenario-specific, which limits their broader applicability.
>
> On point 3, I would like to emphasize that BLEU scores are widely recognized as a poor metric for evaluating machine translation quality, because of their high misalignment with human preferences. BLEU ranked 28/32 metrics for machine translation on WMT23. The reliance on lexical overlap for evaluation is fundamentally flawed; the primary reason BLEU has been used since 2002 is the historical lack of better alternatives. However, with the advent of neural-based metrics, especially for high-resource languages (that studied in the paper), we now have more robust and reliable tools for evaluation. This shift in the field is well-established, and I encourage the authors to consider this consensus and align their approach accordingly.
>
> While it is true that many papers at top-tier conferences still report BLEU scores (including many of my own, my papers are typically done for completeness and comparability with prior work rather than as an endorsement of BLEU’s validity). The prevalence of BLEU in **top-tier conferences**does not inherently justify its continued use as the primary metric. Instead, the authors should focus on understanding which metrics are currently considered higher quality for evaluation, rather than **blindly trusting previous top-tier conference papers**. This is a common pitfall for researchers. A lack of understanding of the metrics could stem from oversight or misinterpretation, but I encourage the authors to address this by conducting a more thorough investigation.
>
> Some papers may help the authors understand why BLEU is not an ideal metric. As for ROUGE, I don't know because it is  very rare for MT. Based on what the authors mentioned, only 1 out of 7 papers referenced ROUGE, and it was in the context of self-driving?
>
> [1] Navigating the Metrics Maze: Reconciling Score Magnitudes and Accuracies
> [2] Results of WMT22 Metrics Shared Task: Stop Using BLEU – Neural Metrics Are Better and More Robust
> [3] Results of WMT23 Metrics Shared Task: Metrics Might Be Guilty but References Are Not Innocent
>
> There should be more papers mentioning this point, but it needs me take a while to find them, so I just stop here. Finally, **I strongly encourage the authors to develop a solid understanding of evaluation metrics before attempting to make contributions to the field.** While this feedback may come across as a bit harsh, I believe it is important to emphasize here.
>
> (I believe the authors had a misunderstanding of my initial review. I did not say that you did not use COMET or you should considering use COMET, nor did I overlook its inclusion. My point was that BLEU is a poor metric compared to COMET, and I suggested that you consider alternatives such as BLEURT)

---

> > ### Author Response · Authors · 2024-12-02
> >
> > We appreciate the feedback, though we feel that referring to our approach as **“blindly trusting previous top-tier conference papers”** does not foster constructive and respectful dialogue.
> >
> > ----
> >
> > - "the authors should focus on understanding which metrics are currently considered higher quality for evaluation, rather than blindly trusting previous top-tier conference papers":
> >
> > Thanks for the suggestion but one of the main metrics we report is COMET score, which is one of the highest quality metrics in the papers referenced by the reviewer themselves in their latest response.
> >
> > ---
> >
> > - While we agree that BLEU has limitations and is not the best metric for translation, it remains relevant to our paper as long as there are researchers paying attention to it (which is evident from the recent papers we listed in our previous response). This is because our paper does not try to improve machine translation, or argue which translation model is the best, or endorse a particular metric. Our paper studies scaling laws for downstream metrics. Therefore, exploring both widely used metrics like BLEU (despite their limitations) and emerging metrics like COMET enhances our paper's relevance.
> >
> > ---
> >
> > - In particular, we use BLEU score to also provide a point of comparison and challenge prior assumptions in the field, e.g., exponential relationship between BLEU score
> > and cross-entropy by [Data and parameter scaling laws for neural machine translation](https://aclanthology.org/2021.emnlp-main.478/). See lines 472-482 and Figure 5 of our paper for details. **We would not be able to challenge this prior assumption without reporting BLEU scores.** The reviewer stated that they themselves also include BLEU score in their papers for similar reasons: "While it is true that many papers at top-tier conferences still report BLEU scores (including many of my own, my papers are typically done for completeness and comparability with prior work rather than as an endorsement of BLEU’s validity)."
> >
> > ----
> >
> > - We also recognize the value of BLEURT, but given the time constraints, we believe that the inclusion of COMET, BLEU, and ROUGE already provides a comprehensive evaluation. **We note that all three references shared by the reviewer in their previous response recommend using COMET score and we included COMET results.**
> >
> > ----
> >
> > - “Findings appear to be overly scenario-specific”:
> >
> > We respectfully disagree with the reviewer. We analyzed many corner cases in a controlled way to isolate the effect of alignment on the value of pretraining in a way that was never done before. We discovered empirical evidence breaking prior assumptions such as the exponential relation between BLEU and cross entropy [1] and cross entropy scaling laws being sufficient for downstream performance [2]. If anything, our paper corrects the misunderstandings of prior papers that made assumptions over not diverse enough settings – missing the corner cases we studied.
> >
> > In case the reviewer is referring to the following sentence in our rebuttal, “We empirically observed that, in the particular setup we considered, as the finetuning data size grows, pretraining brings less value.”, here, we were actually showing a counter example against the reviewer’s quote from another paper [3]  “you only need a large amount of data for pre-training and a small number of data for continued fine-tuning”. Our empirical findings indicate that the quoted suggestion holds only when finetuning data is not sufficiently large (which is a common case in practice). **This shows that this quoted suggestion does not apply to the broader scenarios we studied in our paper.**
> >
> > All in all, our controlled experiment setup allowed us to challenge many common prior assumptions, shed light into the behavior of downstream metrics, and connected these findings to the value of pretraining under scenarios that was not analyzed in a controlled way by prior work.
> >
> > ----
> >
> > We hope this response clarifies our position and encourages a more productive and respectful exchange moving forward.
> >
> > [1] [Data and parameter scaling laws for neural machine translation](https://aclanthology.org/2021.emnlp-main.478/)
> >
> > [2] [Scaling Laws for Transfer](https://arxiv.org/pdf/2102.01293)
> >
> > [3] [A Paradigm Shift in Machine Translation: Boosting Translation Performance of Large Language Models](https://arxiv.org/abs/2309.11674)

---

### Author Response · Authors · 2024-11-27

We thank the reviewers for their supportive and constructive comments. In particular, the reviewers praised the **extensive, interesting, and important experimental results** (Reviewers 2Keq and nK1N), highlighted that **our experiments are controlled and successfully isolate different variables’ effects** (Reviewers tpup and 9RdU), found the **results relevant to a large community** (Reviewer nK1N), liked the **clear presentation and effective structure** (Reviewer nK1N and 9RdU), indicated the **importance of challenging common assumptions about cross entropy and translation quality metrics**.  We revised our paper following the suggestions (the changes are highlighted in blue) and address the reviewers’ questions below individually.

---

### Meta-Review · Area_Chair_o7Yw · 2024-12-26

**Metareview:**

This paper investigates the scaling laws of the MT downstream performance. More importantly, when pretraining and downstream data are well-aligned, both downstream CE loss and translation quality metrics follow predictable scaling laws; when the distributions are misaligned, the CE loss continues to scale monotonically, but translation quality metrics deviate, sometimes worsening with more pretraining data. A novel logarithmic law describes the scaling behavior of translation metrics, which complements prior scaling laws based on CE loss

**Strengths** (1) Comprehensive empirical analysis, (2) reasonable but solid insights on the MT scaling laws based on the data alignment; (3) a new log-law for downstream metric scaling

**Weaknesses** (1) Some reviewers mentioned that "alignment" between pretraining and downstream data is insufficiently defined and unclear how to quantify (addressed during the discussion phase); (2) The study lacks experiments on low-resource, multilingual and cross-domain scenarios; (3) Reviewer argue the conclusion of the paper might be inaccurate as the paper focuses on traditional encoder-decoder architectures, while the mainstream SoTA MT research is shifting to decoder-only models; (4) Additional neural-based metrics can be added as supplement.

**Decision**
This paper makes valuable contributions by exploring scaling laws for MT and proposing a new log law for downstream performance. With the additional improvements, including the definition of alignment, I am leaning toward acceptance.

It is also noted that part of the review (including the discussion) from 2Keq is a bit irrelevant from the focus of the paper. Therefore, I down-weigh the score from 2Keq.

**Additional Comments On Reviewer Discussion:**

The discussion except 2Keq is mainly focusing on the definition of alignment, which the authors addressed with a clear equation.

The discussion between 2Keq leads to a major disagreement on the choice of metrics and the problem setups (encoder-decoder or decoder-only). While the authors provided explanations with existing references, 2Keq insisted on the suggestion of replacing BLEU with neural-based metrics. **After checking the discussion, I believe this discussion diverged from the motivation/conclusion of the paper, and many tones were not appropriate and not constructive.**

---

### Decision · Program_Chairs · 2025-01-22

Accept (Poster)